# Pyk2 modulates hippocampal excitatory synapses and contributes to cognitive deficits in a Huntington's disease model

Albert Giralt[1,2,3], Veronica Brito[4,5,6,7], Quentin Chevy[1,2,3,†], Clémence Simonnet[1,2,3], Yo Otsu[1,2,3], Carmen Cifuentes-Díaz[1,2,3], Benoit de Pins[1,2,3], Renata Coura[1,2,3], Jordi Alberch[4,5,6,7], Sílvia Ginés[4,5,6,7], Jean-Christophe Poncer[1,2,3] & Jean-Antoine Girault[1,2,3]

The structure and function of spines and excitatory synapses are under the dynamic control of multiple signalling networks. Although tyrosine phosphorylation is involved, its regulation and importance are not well understood. Here we study the role of Pyk2, a non-receptor calcium-dependent protein-tyrosine kinase highly expressed in the hippocampus. Hippocampal-related learning and CA1 long-term potentiation are severely impaired in Pyk2-deficient mice and are associated with alterations in NMDA receptors, PSD-95 and dendritic spines. In cultured hippocampal neurons, Pyk2 has autophosphorylation-dependent and -independent roles in determining PSD-95 enrichment and spines density. Pyk2 levels are decreased in the hippocampus of individuals with Huntington and in the R6/1 mouse model of the disease. Normalizing Pyk2 levels in the hippocampus of R6/1 mice rescues memory deficits, spines pathology and PSD-95 localization. Our results reveal a role for Pyk2 in spine structure and synaptic function, and suggest that its deficit contributes to Huntington's disease cognitive impairments.

[1] Inserm UMR-S 839, 75005 Paris, France. [2] Université Pierre & Marie Curie, Sorbonne Universités, 75005 Paris, France. [3] Institut du Fer a Moulin, 75005 Paris, France. [4] Departament de Biomedicina, Facultat de Medicina, Universitat de Barcelona, 08036 Barcelona, Spain. [5] Institut d'Investigacions Biomèdiques August Pii Sunyer (IDIBAPS), 08036 Barcelona, Spain. [6] Centro de Investigación Biomédica en Red Sobre Enfermedades Neurodegenerativas (CIBERNED), 28031 Madrid, Spain. [7] Institut de Neurociencies, Universitat de Barcelona, 08036 Barcelona, Spain. † Present address: Cold Spring Harbor Laboratory, 1 Bungtown Road, Cold Spring Harbor, New York 11724, USA. Correspondence and requests for materials should be addressed to J.-A.G. (email: jean-antoine.girault@inserm.fr).

Synaptic function and plasticity, as well as spine morphology are regulated by multiple signalling pathways that integrate the diversity of signals converging on synapses. Receptors for neurotransmitters, as well as numerous other post-synaptic proteins are phosphorylated by a variety of serine/threonine protein kinases, many of which have been extensively investigated, especially in the context of synaptic plasticity. Tyrosine kinases have also been reported to contribute to the regulation of post-synaptic proteins, including regulation of NMDA (N-methyl-D-aspartate) glutamate receptors by Src family kinases (SFKs) and proline-rich tyrosine kinase 2 (Pyk2)[1,2]. However, the functional importance of these regulations in vivo is unknown.

Pyk2 is a non-receptor tyrosine kinase that can be activated by $Ca^{2+}$ and is highly expressed in forebrain neurons, especially in the hippocampus[3,4]. Previous findings indicated a role for Pyk2 in synaptic plasticity[1,5–7] and its gene, PTK2B, is a susceptibility locus for Alzheimer's disease[8]. Pyk2 is activated by $Ca^{2+}$, and although the mechanism has not been fully elucidated, it probably involves dimer assembly[9], which triggers its autophosphorylation at Tyr402 and the recruitment of SFKs[10]. Tyr402 phosphorylation is increased by neuronal depolarization[11] and tetanic stimulation[7] in hippocampal slices, and by activation of NMDA[5] or group I metabotropic[12] glutamate receptors in cultured hippocampal neurons. Pyk2 and SFKs are part of the NMDA receptor complex[7,13] and Pyk2 interacts directly with post-synaptic density (PSD) proteins PSD95 (ref. 14), SAP102 (ref. 14) and SAPAP3 (ref. 15). Long-term potentiation (LTP) of CA1 synapses requires protein tyrosine phosphorylation[16,17] and is prevented by a kinase-dead Pyk2 (ref. 7) or by competition of Pyk2:PSD95 interaction[5]. These results led to the suggestion of a role of $Ca^{2+}$-induced activation of Pyk2 in regulating NMDA receptor function and synaptic plasticity, likely through recruitment of SFKs[1,2]. However, the functional relevance of these findings in vivo is not known and the role of Pyk2 in hippocampal physiology or pathology has not been investigated. Pyk2 knockout mice display a mild immunological phenotype but their nervous system has not been studied[18].

Here we show that the inactivation of one or two alleles of the Ptk2b gene in mice does not alter hippocampal development but prevents hippocampal-dependent memory tasks and LTP. We provide evidence for multiple roles of Pyk2 in spine morphology and post-synaptic structure. Moreover, we show that Pyk2 is decreased in the hippocampus of patients with Huntington's disease (HD), an inherited neurodegenerative disorder, which results from the expansion of a CAG trinucleotide repeat in the huntingtin (Htt) gene[19]. Pyk2 is also decreased in R6/1 mice, which express a mutated form of Htt, and these mice display a hippocampal phenotype similar to that observed in Pyk2 mutant mice. This phenotype is partly rescued by restoring Pyk2 levels in R6/1 mice, suggesting a reversible role of Pyk2 deficit in the HD mouse model.

## Results

**Pyk2 knockout impairs hippocampal-dependent memory and LTP.** To study the role of Pyk2 in the brain, we used a knockout mouse line[20] that we recently generated. As previously observed for a similar line[18], these mice bred normally and there were no differences between Pyk2$^{+/+}$, Pyk2$^{+/-}$ and Pyk2$^{-/-}$ mice in body weight, muscular strength, general locomotor activity or anxiety levels evaluated in the elevated plus-maze (Supplementary Fig. 1a–d). We tested Pyk2$^{+/+}$, Pyk2$^{+/-}$ and Pyk2$^{-/-}$ littermate mice in two simple tasks that depend on hippocampus-mediated spatial memory[21,22]. In the Y-maze spontaneous alternation task, Pyk2$^{+/+}$ mice showed a significant preference for the new arm 2 h after exposure to the other arm, whereas both Pyk2$^{+/-}$ and Pyk2$^{-/-}$ littermates

explored equally both arms (Fig. 1a). In the novel object location (NOL) test, 24 h after a first exposure, wild-type mice spent more time exploring the object placed at a new location (Fig. 1b). In contrast, both Pyk2$^{+/-}$ and Pyk2$^{-/-}$ mice did not display any preference for either object (Fig. 1b). These results revealed spatial memory deficits in both heterozygous and homozygous mutant mice.

We next examined whether these behavioural deficits were accompanied by altered synaptic plasticity in hippocampal slices. We restricted our study to CA1, a hippocampal region extensively implicated in spatial learning. High-frequency conditioning tetanus of Schaffer collaterals ($5 \times 1$ s at 100 Hz) induced LTP in CA1 of wild-type mice (Fig. 1c). In contrast, no LTP was observed in slices from Pyk2$^{+/-}$ or Pyk2$^{-/-}$ mice (Fig. 1c,d). We also examined a form of short-term plasticity at the same synapses. Paired-pulse facilitation was observed in wild-type mice but was markedly decreased in both homozygous and heterozygous Pyk2 mutant mice (Fig. 1e and Supplementary Fig. 1e), suggesting the existence of a presynaptic role of Pyk2. Taken together, these results show that deletion of Pyk2 impairs hippocampus-dependent memory and synaptic plasticity in CA1. Importantly, the heterozygous mutation of Pyk2 was as severe as the full deletion, indicating that Pyk2 levels may be limiting for hippocampal plasticity.

**Alteration in NMDA receptors and PSD-95 in Pyk2 mutant mice.** To explore the molecular consequences of Pyk2 deficit, we examined the levels of proteins previously associated with the Pyk2 pathway at synapses by immunoblotting. In hippocampal tissue of Pyk2$^{+/-}$ mice, Pyk2 protein was decreased by about 50% as compared to wild-type littermates, and was not detectable in Pyk2$^{-/-}$ mice (Fig. 2a,b). No N-terminal truncated fragment was detected in the knockout mice (Supplementary Fig. 2a), showing that deletion of exons 15–18 in the Pyk2 gene[20] destabilized the resulting mRNA and/or protein. There was no compensatory alteration of the related focal adhesion kinase (FAK, Fig. 2a,b). In both Pyk2$^{+/-}$ and Pyk2$^{-/-}$ mice, the activated form of SFKs (pY-SFK, pTyr420 in Fyn) was markedly reduced, whereas Fyn levels were unchanged (Fig. 2a,b), underlining the contribution of Pyk2 in regulating SFKs phosphorylation. In contrast, there was no change in the basal phosphorylation (activation) of ERK1/2 (Supplementary Fig. 2b,c), reported to be downstream of Pyk2 in some cell systems[10], including in hippocampal neurons in culture[6], but not in adult slices[23]. We then focused on glutamate receptors. We found no consistent change between genotypes in GluA1 and GluA2 AMPA receptors subunits, or in their phosphorylated forms pSer831-GluA1 and pTyr876-GluA2 (Supplementary Fig. 2d,e). NMDA receptors GluN1 levels were not changed either (Fig. 2d,e). In contrast, we observed marked alterations of NMDA receptor N2 subunits. The phosphorylated forms of GluN2A and GluN2B, pTyr1246- and pTyr1325-GluN2A, and pTyr1472-GluN2B were decreased in Pyk2$^{-/-}$ compared to wild-type mice. Total GluN2B was not changed indicating deficient tyrosine phosphorylation in the absence of Pyk2 (Fig. 2d,e). In contrast, total GluN2A was decreased (Fig. 2d,e). We also examined PSD-95, a post-synaptic scaffolding protein that interacts with both NMDA receptors and Pyk2 (refs 5,14). The levels of PSD-95 were markedly decreased in homozygous mutant mice (Fig. 2d,e). Thus, in contrast with the behavioural and physiological deficits, which appeared as pronounced in heterozygous as in homozygous mutant mice (see Fig. 1), the protein alterations in Pyk2$^{+/-}$ mice were intermediate between wild type and Pyk2$^{-/-}$, indicating some proportionality between the decrease in Pyk2 and its consequences on other proteins. To determine the changes in receptors that took place at

synapses, we carried out subcellular fractionation and isolated postsynaptic densities (PSDs). The amounts of GluN1, GluN2A, GluN2B and PSD-95 were decreased in the PSD fraction of Pyk2$^{-/-}$ mice as compared to wild type (Fig. 2e,f). These results showed that the lack of Pyk2 signalling resulted in decreased tyrosine phosphorylation of SFKs and GluN2B subunits as well as decreased levels of GluN2A and PSD95 total protein. The enrichment of all these proteins in PSDs was markedly decreased, indicating a key role of Pyk2 in regulating the recruitment of post-synaptic proteins to PSDs.

**Spines are altered in the hippocampus of Pyk2 mutant mice.** To explore how Pyk2 deficit could induce alterations of synaptic proteins, we first determined its localization in CA1. Pyk2 immunofluorescence in the neuropil was punctate and appeared to surround MAP2-positive dendritic processes (Fig. 3a). Some Pyk2-positive puncta co-localized with PSD-95-positive puncta, identifying them as PSDs (Fig. 3b). We then examined Pyk2 immunoreactivity by electron microscopy. Pyk2-positive immunogold particles were found in both presynaptic elements and dendritic spines (Fig. 3c and Supplementary Fig. 3a). Pyk2 was enriched in asymmetric (presumably excitatory) synapses as compared to symmetric (presumably inhibitory) synapses (Supplementary Fig. 3a–d). Because of Pyk2 co-localization with PSD-95 and of the marked decrease in PSD-95 in Pyk2$^{-/-}$ mice, we quantified PSD-95-positive puncta in CA1 *stratum radiatum* of wild-type and mutant mice. The number of PSD-95-positive puncta was significantly reduced in Pyk2$^{+/-}$ and even more so in Pyk2$^{-/-}$ as compared to Pyk2$^{+/+}$ mice (Fig. 3d,e).

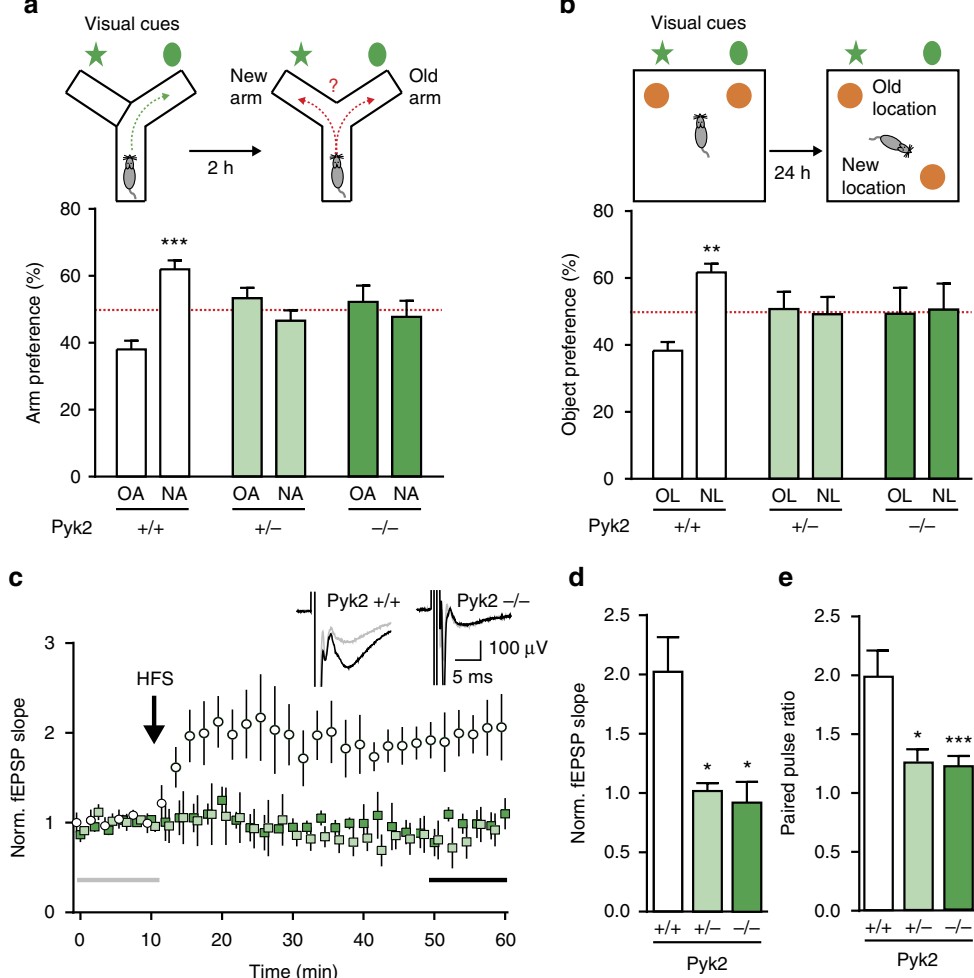

**Figure 1 | Spatial learning and memory and CA1 LTP deficits in Pyk2 mutant mice. (a)** In the spontaneous alternation test, Pyk2$^{+/+}$, Pyk2$^{+/-}$ and Pyk2$^{-/-}$ 3-month-old mice were placed for 10 min in a Y-maze with one arm closed (upper left panel). Two hours later, they were put in the same maze with the new arm (NA) open and the percentage of time exploring the NA and the previously explored (old arm, OA) was compared (upper right panel). Two-way ANOVA interaction $F_{(2,48)} = 11.6$, $P < 0.0001$, OA versus NA Holm-Sidak's test, Pyk2$^{+/+}$, $t = 4.6$, $P < 0.0001$, Pyk2$^{+/-}$, $t = 1.58$, Pyk2$^{-/-}$, $t = 0.81$. **(b)** In the NOL test, the percentage of time exploring the displaced object (new location, NL, 24 h after first exposure) and the unmoved object (old location, OL) was compared (upper panels). Two-way ANOVA interaction $F_{(2,50)} = 3.41$, $P = 0.041$, OL versus NL Holm-Sidak's test, Pyk2$^{+/+}$, $t = 3.1$, $P < 0.01$, Pyk2$^{+/-}$, $t = 0.23$, Pyk2$^{-/-}$, $t = 0.14$. In **a,b**, 7–12 mice were used per genotype; the red dotted line indicates the chance level. **(c,d)** Schaffer collaterals were stimulated in hippocampal slices (one–three slices per animal) from 3-4-week-old Pyk2$^{+/+}$ ($n = 5$), Pyk2$^{+/-}$ ($n = 6$) and Pyk2$^{-/-}$ ($n = 4$) mice, and fEPSP were recorded in CA1, before and after HFS ($5 \times 1$ s at 100 Hz). **(c)** Time course of fEPSP slope. Insets show typical traces before (grey) and 40 min after (black) HFS in Pyk2$^{+/+}$ and Pyk2$^{-/-}$ slices. **(d)** Ten-min average of fEPSP slope 40 min after HFS, normalized to the mean of 10-min baseline (corresponding time points are indicated in **c** by grey and black horizontal lines). Kruskal–Wallis = 9.37, $P = 0.0024$, *post hoc* analysis with Dunn's multiple comparisons test. **(e)** Paired-pulse ratio (50-ms interval, see Supplementary Fig. 1e) at the same synapses. $n = 3$–5 mice per group, two–four slices per mouse. Kruskal–Wallis = 15.62, $P = 0.0004$. In **a–e**, values are means + s.e.m., *$P < 0.05$, **$P < 0.01$, ***$P < 0.001$.

This effect appeared consistent throughout CA1 depth (Supplementary Fig. 3e).

To determine the consequences of these alterations on spines, we analysed spine density and morphology in CA1 pyramidal neurons, using the Golgi-Cox method (Fig. 3f). The apical dendritic spines density was decreased in Pyk2$^{+/-}$ ($-8\%$) and Pyk2$^{-/-}$ ($-16\%$) mice as compared to wild type (Fig. 3g). The decrease in spine number was less pronounced than the decrease in PSD-95 puncta, possibly due to an immunofluorescence detection threshold and/or an increased number of spines lacking PSD-95. To determine whether the absence of Pyk2 also affected spine morphology, we quantified the spine head diameter and spine neck length. Spine head size did not change between genotypes (Fig. 3h), whereas spine neck length was decreased in

Pyk2 mutant mice (Fig. 3i). Altogether, these data show that the lack of Pyk2 leads to a decrease in PSD-95 at synapses and a decreased number of PSDs and spines.

**Adult hippocampal Pyk2 deletion recapitulates the phenotype.** Although Pyk2 expression in the hippocampus is mostly post-natal[4], the severe alterations observed in Pyk2$^{+/-}$ and Pyk2$^{-/-}$ mice could result from developmental effects. To rule out this possibility, we used 3-month-old mice bearing floxed Pyk2 alleles (Pyk2$^{f/f}$ mice). Mice received a bilateral stereotaxic injection in CA1 of adeno-associated virus expressing Cre recombinase and GFP (AAV-Cre) or expressing GFP alone (AAV-GFP), as a control (Fig. 4a). Three weeks after AAV-Cre injection, Pyk2 expression disappeared in CA1, whereas the injection of

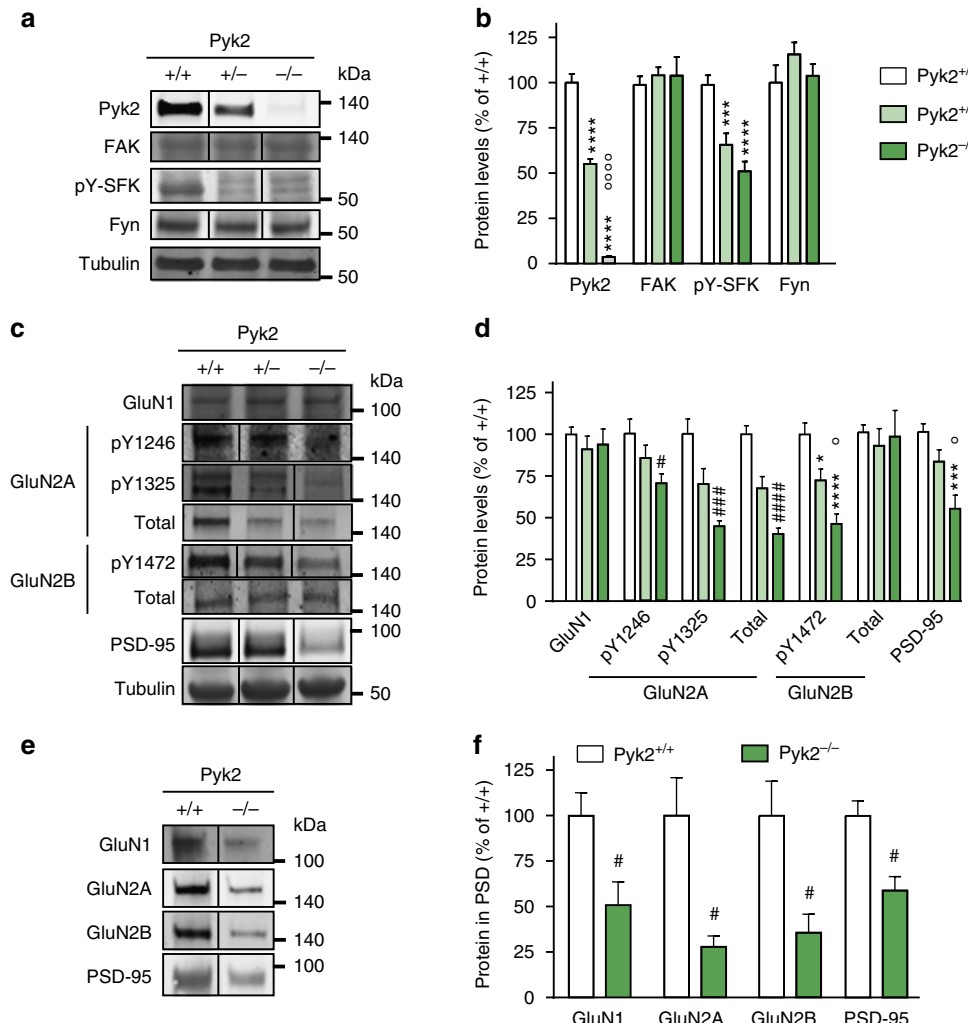

**Figure 2 | Hippocampal proteins phosphorylation and levels in Pyk2-deficient mice.** (**a**) Immunoblotting analysis of Pyk2, the related tyrosine kinase FAK, the active autophosphorylated form of Src-family kinases (pY-SFK, pTyr-420 in Fyn), Fyn and tubulin as a loading control in 3-month-old Pyk2$^{+/+}$, Pyk2$^{+/-}$ and Pyk2$^{-/-}$ littermates. (**b**) Densitometry quantification of results as in **a**. Data were normalized to tubulin for each sample and expressed as percentage of wild type. (**c**) NMDA receptors subunits phosphorylated residues, total levels and PSD-95 were analysed by immunoblotting. (**d**) Results as in **c** were quantified and analysed as indicated in **b**. In **b** and **d**, statistical analysis was done with one-way ANOVA and Holm-Sidak's multiple comparisons test or Kruskal–Wallis and Dunn's test depending on the normality of distribution (see Supplementary Table 1 for tests used, values and number of mice). (**e**) PSD fraction was prepared from hippocampus of Pyk2$^{+/+}$ and Pyk2$^{-/-}$ mice and NMDA receptor subunits and PSD-95 were analysed in this fraction by immunoblotting. (**f**) Quantification of immunoblots as in **e**. Data are expressed as a percentage of the mean values in wild-type PSDs. Two-tailed Mann and Whitney test ($n=7^{+/+}$ and $5^{-/-}$): GluN1, $t_{10}=3.52$, $P=0.0056$, GluN2A, $t_{10}=2.68$, $P=0.023$, GluN2B, $t_{10}=2.69$, $P=0.022$, PSD-95, $t_{10}=2.66$, $P=0.024$. In **a,c,e**, molecular weight markers positions are indicated in kDa. In **b,d**, Holm-Sidak's versus wild type, *$P<0.05$, **$P<0.01$, ***$P<0.001$ and ****$P<10^{-4}$; significant differences between $-/-$ and $-/+$ are indicated with °$P<0.05$, °°$P<0.01$ and °°°°$P<10^{-4}$. In Dunn's test (**d**) and Mann and Whitney's test (**f**), significant differences versus wild type are indicated with # $P<0.05$, ### $P<0.01$ and #### $P<10^{-4}$. In all graphs, data are means + s.e.m. Uncropped blots for **a,c** and **e** are shown in Supplementary Figs 5, 6 and 7, respectively.

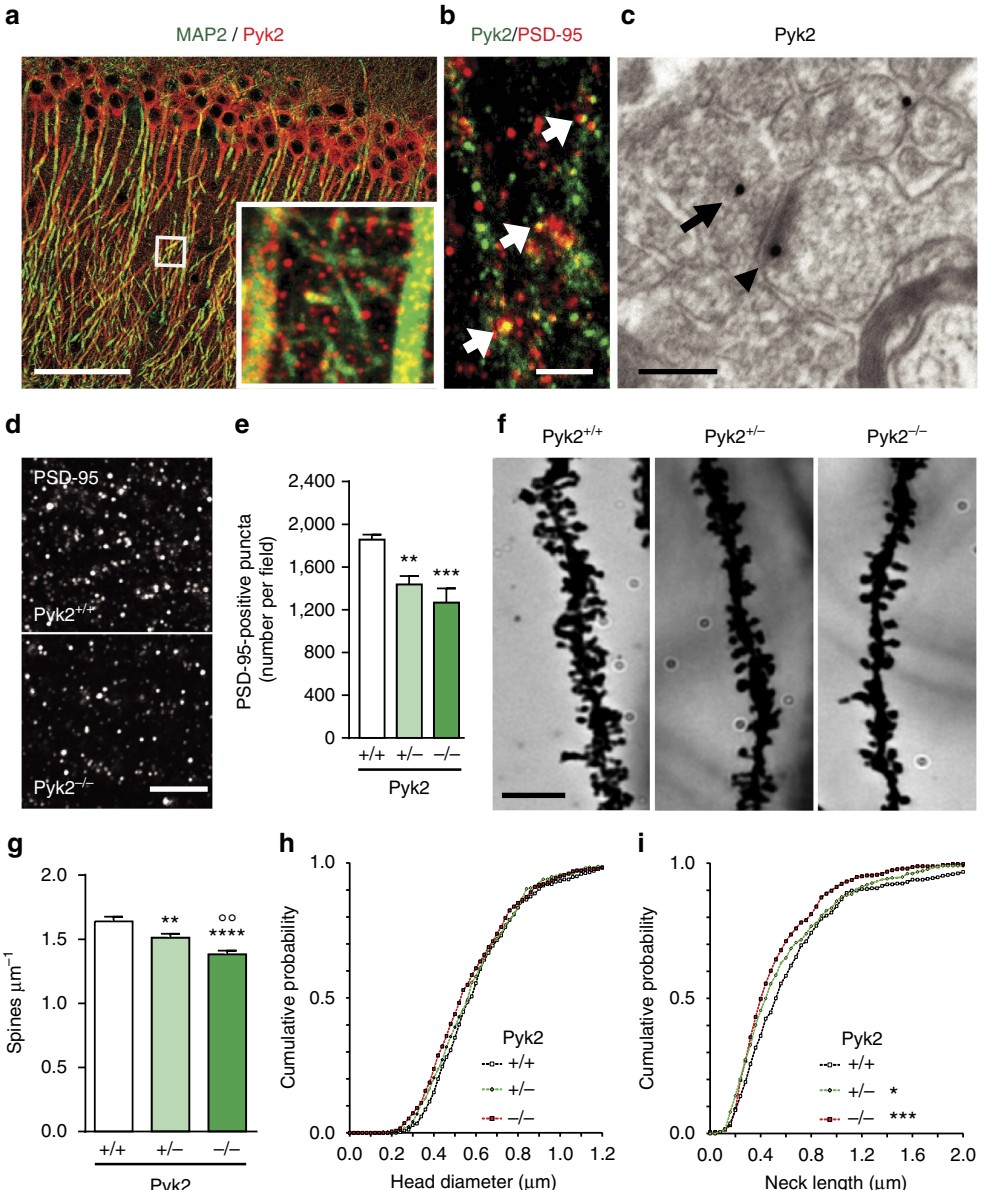

**Figure 3 | Pyk2 localization and dendritic spine density and morphology in Pyk2-deficient mice.** (**a,b**) Confocal microscopy images of CA1 *stratum radiatum* immunostained for (**a**) Pyk2 (red) and MAP2 (green; inset, higher magnification of the indicated white box) and for (**b**) Pyk2 (green) and PSD-95 (red; white arrows, double-labelled puncta). Scale bars, 80 μm (**a**) and 3 μm (**b**). (**c**) Electron microscopy in the same region showing of Pyk2 immunoreactive gold particles in a presynaptic terminal (arrow) and a PSD (arrowhead). Scale bar, 0.2 μm. (**d**) Immunofluorescence PSD-95-positive puncta in the CA1 *stratum radiatum* from Pyk2$^{+/+}$ and Pyk2$^{-/-}$ mice. Scale bar, 5 μm. (**e**) Quantification of puncta as in **d**. Data are means + s.e.m. (7–10 mice per genotype, three quantified sections per mouse). One-way ANOVA $F_{(2,21)} = 10.23$, $P = 0.0008$. Holm-Sidak's multiple comparisons test versus +/+, **$P < 0.01$, ***$P < 0.001$. (**f**) Golgi-Cox-stained apical dendrites of CA1 *stratum radiatum* pyramidal neuron from Pyk2$^{+/+}$, Pyk2$^{+/-}$ and Pyk2$^{-/-}$ mice. Scale bar, 3 μm. (**g**) Quantification of spine density in dendrites as in **f**, three–four animals per genotype, one-way ANOVA, $F_{(2,146)} = 14.95$, $P < 10^{-4}$ ($n = 47$–$54$ dendrites per group), *post hoc* analysis with Holm-Sidak's multiple comparisons test versus +/+, **$P < 0.01$, ****$P < 10^{-4}$ and $-/-$ versus $-/+$, °°$P < 0.01$. (**h,i**) Cumulative probability of spine head diameter (**h**, $n = 80$) and spine neck length (**i**, $n = 115$) in ∼60 dendrites from three–four animals per genotype. Distributions were compared with the Kolmogorov–Smirnov test: spine head diameter no significant difference, neck length +/+ versus +/−, $D = 0.108$, $P = 0.04$, +/+ versus $-/-$, $D = 0.154$, $P = 0.0005$. In **e,g**, data are means + s.e.m. All mice were 3–4-month old.

AAV-GFP had no effect (Fig. 4a). In the NOL test, AAV-GFP-injected mice showed increased preference for the object placed at the new location, whereas AAV-Cre-injected mice did not (Fig. 4b). We analysed spine density in CA1 apical dendrites and found it was significantly reduced in AAV-Cre-injected mice as compared to AAV-GFP-injected mice (Fig. 4c,d). We also quantified a reduced number of PSD-95-positive puncta in CA1 *stratum radiatum* of AAV-Cre-injected mice as compared to AAV-GFP-injected mice (Fig. 4e,f). Altogether these results show that local deletion of Pyk2 in CA1 of adult mice recapitulates behavioural and morphological deficits observed in Pyk2$^{+/-}$ and Pyk2$^{-/-}$ mice, ruling out a developmental effect in the phenotype of Pyk2 mutant mice.

**Pyk2 is needed for NMDA-induced PSD-95 recruitment in spines.** PSD-95 undergoes rapid activity-dependent relocalization[24].

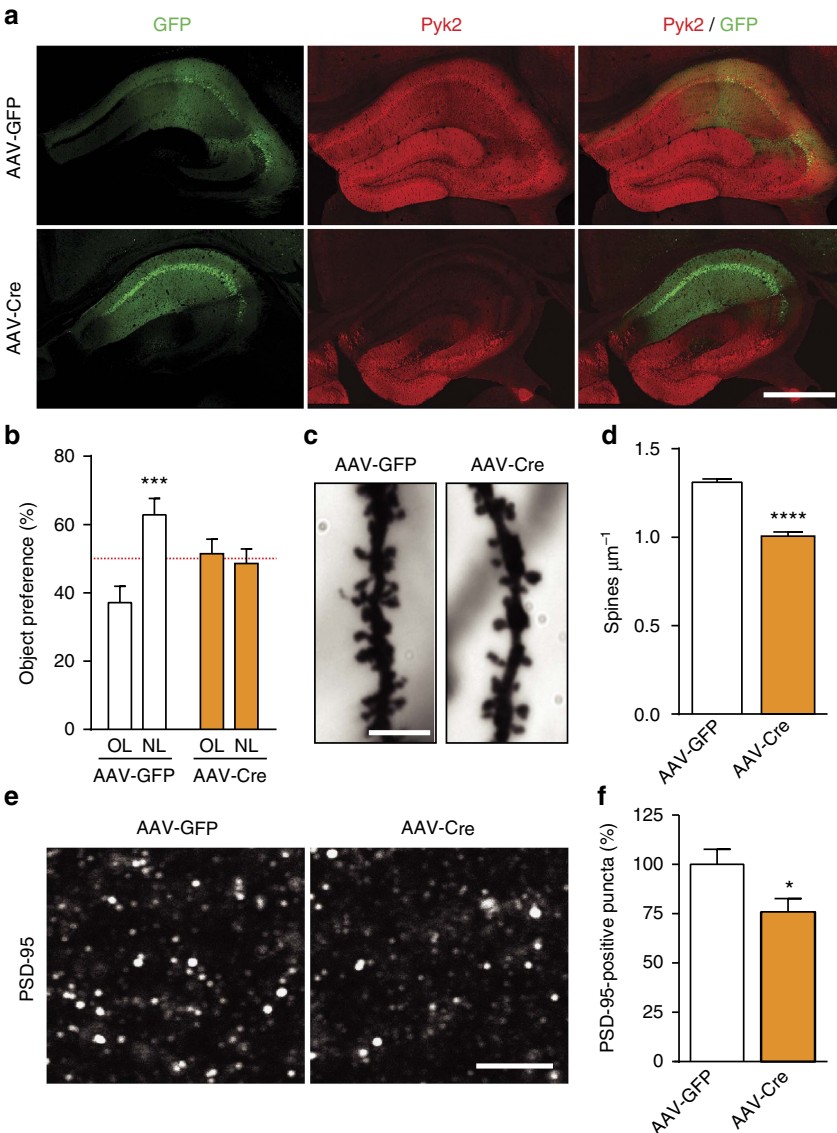

**Figure 4 | Pyk2 ablation in CA1 from adult mice induces spatial learning deficits and spine alterations.** (**a**) Mice with floxed Pyk2 alleles (Pyk2[f/f], 4-week-old) were bilaterally injected in dorsal hippocampus CA1 with AAV expressing GFP (AAV-GFP) or GFP-Cre (AAV-Cre). GFP fluorescence (green) and Pyk2 immunoreactivity (red) were detected with confocal microscope (stitched pictures). With both viruses widespread, GFP expression is present in CA1 and Pyk2 is reduced in CA1 of AAV-Cre-injected mice. Scale bar, 200 μm. (**b**) AAV-GFP and AAV-Cre mice were subjected to the NOL test as in Fig. 1b and the percentage of time exploring the displaced object (NL) compared to that exploring the unmoved object (OL). Two-way ANOVA interaction $F_{(1,44)} = 9.94$, $P = 0.003$, OL versus NL Holm-Sidak's test, AAV-GFP, $t = 4.0$, $P < 0.001$, AAV-Cre, $t = 0.45$, ns (12 mice per group). The red dotted line indicates the chance level. (**c**) Representative Golgi-Cox-stained apical dendrites from CA1 pyramidal neurons of AAV-GFP and AAV-Cre mice. Scale bar, 4 μm. (**d**) Quantification of spine density in dendrites stained as in **c**, 81–86 dendrites from four mice per genotype. Student's $t$-test $t_{165} = 10.1$, $P < 10^{-4}$. (**e**) PSD-95 immunoreactive puncta in CA1 *stratum radiatum* of AAV-GFP and AAV-Cre mice. Scale bar, 4 μm (**c,e**). (**f**) Quantification of PSD-95-positive puncta density as in **e**, three sections per mouse, six–eight mice per genotype, Student's $t$-test $t_{12} = 2.36$, $P < 0.5$. In **a,d,f**, data are means + s.e.m., $*P < 0.05$, $***P < 0.001$ and $****P < 10^{-4}$.

Although neuronal stimulation decreases PSD-95 palmitoylation-dependent synaptic targeting[25] and increases its ubiquitination and degradation[26], prolonged neuronal activity was shown to increase PSD-95 synaptic concentration[27]. PSD-95 is phosphorylated on multiple tyrosine residues and this phosphorylation can increase its synaptic clustering[28,29]. Since PSD-95 synaptic clustering was decreased in the absence of Pyk2 (Figs 3d,e and 4e,f), we hypothesized Pyk2 may influence the synaptic localization of PSD-95. We tested this hypothesis using hippocampal neurons in primary culture at ∼21–22 DIV. As expected, glutamate treatment (40 μM, 15 min) increased Pyk2

phosphorylation at Tyr402 in hippocampal neurons in culture and this effect was prevented by an NMDA receptor antagonist, MK801 (10 μM, Fig. 5a,b). The size of PSD-95-positive puncta measured 3 h after glutamate treatment was increased and this effect was also prevented by MK801 (Fig. 5c,d). We then compared the effects of glutamate on the size of PSD-95 puncta in neurons from wild-type and Pyk2-KO mice (Fig. 5e,f). In the absence of Pyk2, the effects of glutamate on the size of PSD-95 puncta were lost (Fig. 5e,f). Taken together, these results reveal a role for Pyk2 in NMDA receptor-induced PSD-95 recruitment at post-synaptic sites.

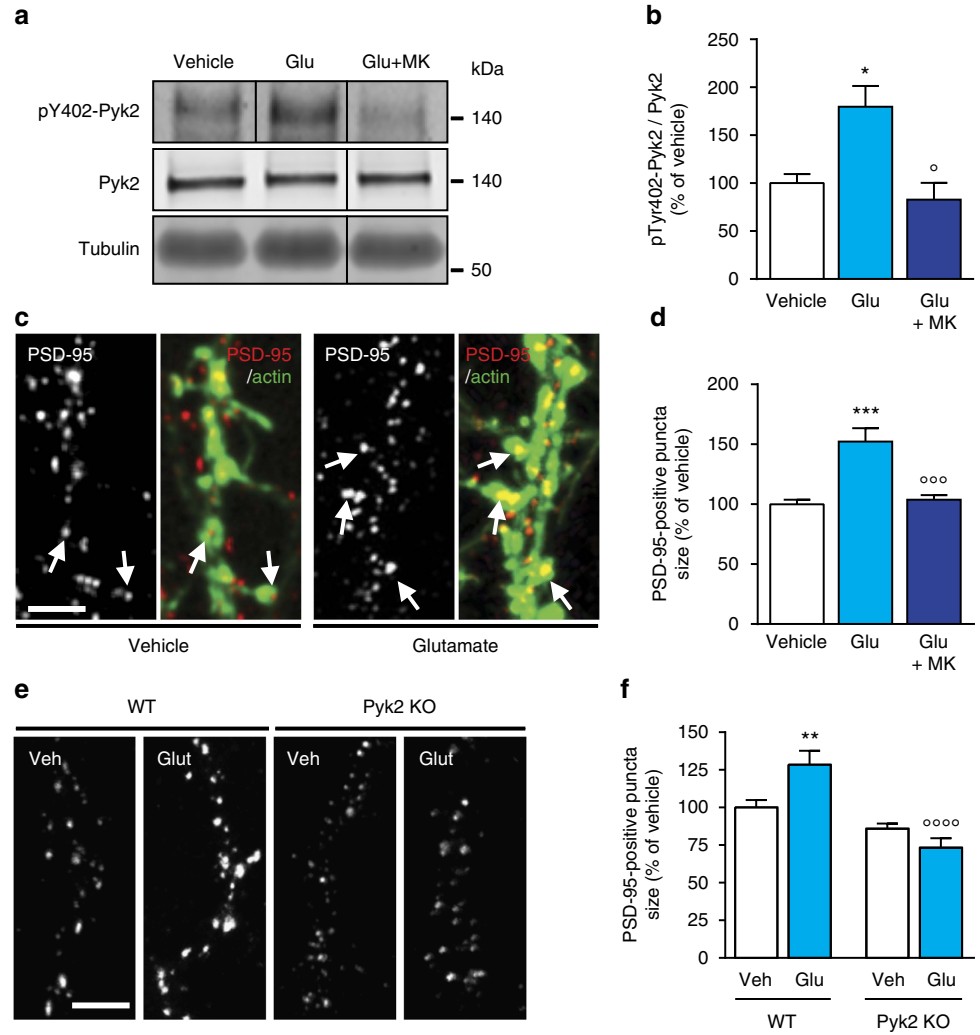

**Figure 5 | Pyk2 modulates glutamate-induced PSD-95 accumulation in dendritic spines.** (**a**) Hippocampal neurons were cultured for 3 weeks and treated for 15 min with vehicle or glutamate (Glu, 40 μM) without or with MK801 (MK, 10 μM), added 30 min before. PhosphoTyr402-Pyk2 (pY402-Pyk2), Pyk2 and α-tubulin as a loading control were analysed by immunoblotting. Molecular weight markers position is indicated in kDa. (**b**) Densitometric quantification of results as in **a**. One-way ANOVA ($F_{(2,13)} = 8.02$, $P = 0.005$, $n = 4$–7 per group) and *post hoc* Holm-Sidak's test for multiple comparisons. (**c**) Cultured hippocampal neurons were treated with vehicle or glutamate (40 μM) without or with MK801 (10 μM) for 3 h, fixed and labelled for PSD-95 immunoreactivity and rhodamine–phalloidin (an F-actin marker) to identify PSD-95-positive puncta localized in dendritic spines (arrows). (**d**) The size of these PSD-95-positive puncta was measured and analysed with one-way ANOVA ($F_{(2,30)} = 15.37$, $P < 0.0001$, $n = 10$–12 per group) and Holm-Sidak's test. (**e**) Hippocampal neurons from wild-type (WT) or Pyk2 KO mice were treated for 3 h with vehicle (Veh) or glutamate (40 μM) and immunostained for PSD-95. (**f**) The size of spine-associated PSD-95-positive puncta was measured in Pyk2$^{+/+}$ and Pyk2$^{-/-}$ hippocampal cultures treated as in **e** and quantified ($n = 18$–27 per group). Statistical analysis with two-way ANOVA (interaction $F_{(1,89)} = 12.42$, $P = 0.0007$, glutamate effect, $F_{(1,89)} = 1.84$, $P = 0.18$, genotype effect, $F_{(1,89)} = 35.29$, $P < 10^{-4}$) and *post hoc* multiple comparisons Holm-Sidak's test. In **d,f**, one–two dendrites per neuron from two to three independent experiments were measured. In **b,d,f**, data are means + s.e.m., *$P < 0.05$, **$P < 0.01$, ***$P < 0.001$, as compared to vehicle-treated Pyk2$^{+/+}$ cultures; °$P < 0.05$, °°°$P < 0.001$ and °°°°$P < 10^{-4}$, as compared to glutamate-treated Pyk2$^{+/+}$ cultures. Scale bars, 5 μm (**c** and **e**). Uncropped blots for **a** are shown in Supplementary Fig. 8.

**Pyk2 function in spines is partly phosphorylation dependent.** Since Pyk2 is a large protein that has tyrosine kinase activity, including functionally important autophosphorylation activity, and interactions with multiple partners[9], we examined which of its molecular properties were required for regulation of PSD-95 and spines. We transfected wild-type and Pyk2-KO hippocampal cultures with GFP or GFP fused to either Pyk2, or Pyk2$_{1-840}$, unable to bind to PSD-95 (ref. 5), or Pyk2$_{Y402F}$ with a point mutation of the autophosphorylation site or to kinase-dead Pyk2 (Pyk2-KD) with a K457A mutation[30]. We first analysed the size of PSD-95-positive puncta in these various conditions (Fig. 6a). As in untransfected neurons (see Fig. 5c–f), glutamate treatment increased the size of PSD-95-positive puncta in wild-type cultures

transfected with either GFP or GFP:Pyk2, used as controls (Fig. 6a,b). Glutamate effects were absent in KO cultures transfected with GFP, but were rescued by Pyk2:GFP transfection (Fig. 6a,b). In contrast, glutamate treatment did not increase PSD-95-positive puncta size in Pyk2$^{-/-}$ cultures transfected with GFP:Pyk2$_{1-840}$, GFP:Pyk2$_{Y402F}$ or GFP:Pyk2-KD (Fig. 6a,b). These results show that the autophosphorylation site, Tyr-402, the kinase activity and the C-terminal domain of Pyk2 are all essential for glutamate-induced PSD-95 synaptic translocation.

We then examined if the spine density could be rescued in cultured hippocampal neurons. Dendritic spine density was reduced in Pyk2-KO neurons as compared to wild type (Fig. 6c,d), as observed *in vivo* (Fig. 3f,g). Transfection of

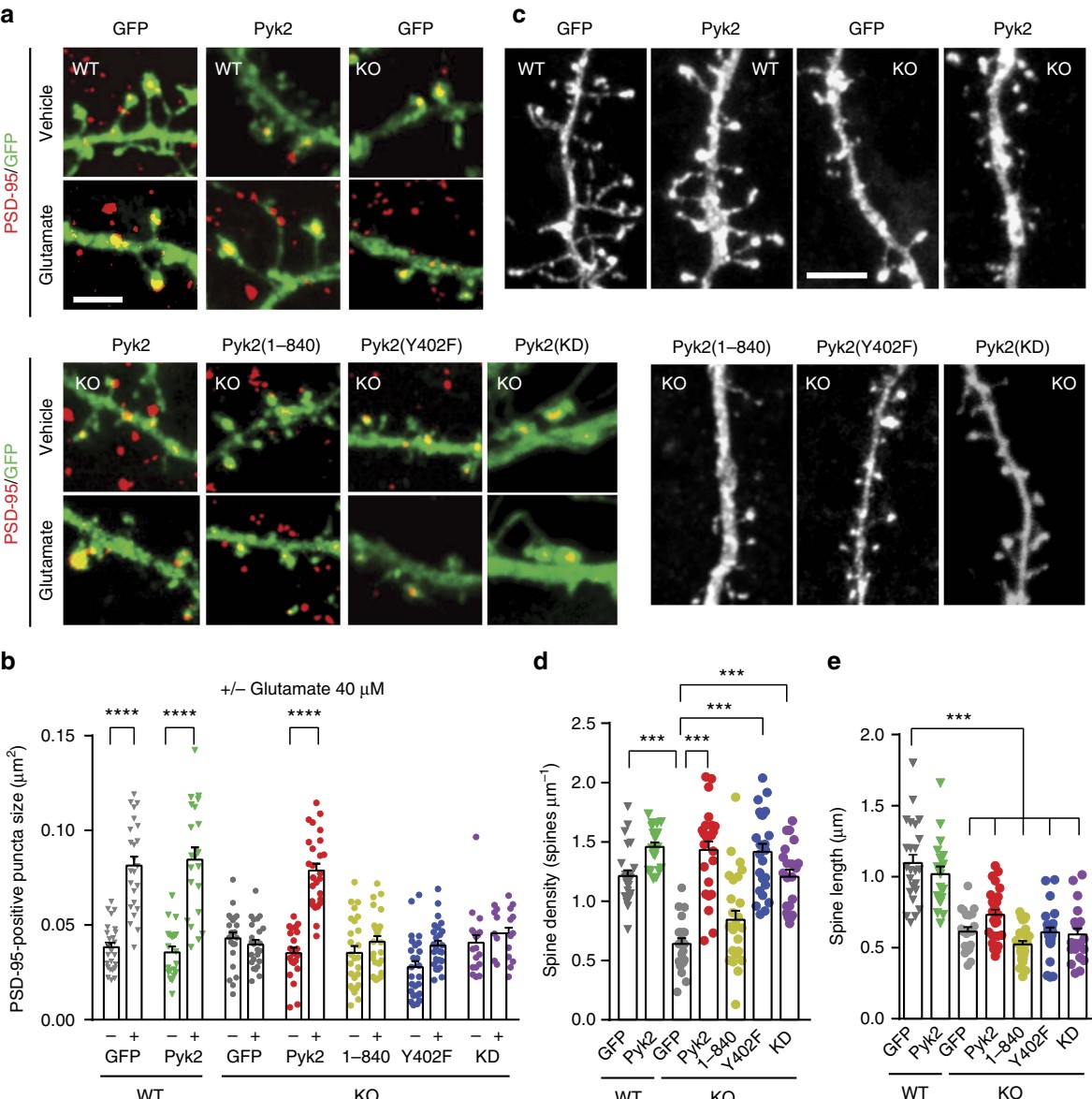

**Figure 6 | Autophosphorylation-dependent and -independent roles of Pyk2 in dendritic spines.** (a) Hippocampal neurons from wild-type (WT) and Pyk2 KO mice were cultured for 21–22 days, transfected with plasmids coding GFP or GFP fused to wild-type Pyk2, to Pyk2(1-840), Pyk2(Y402F) or Pyk2-KD (as indicated), and treated with vehicle or glutamate (Glu, 40 μM, 3 h). Neurons were imaged for GFP fluorescence (green) and PSD-95 immunoreactivity (red). (b) Quantification of GFP/PSD-95 double-positive puncta size (that is, yellow puncta) as in a. Two-way ANOVA: interaction, $F_{(6,312)} = 19.07$, $P < 10^{-4}$, glutamate effect, $F_{(1,312)} = 134.3$, $P < 10^{-4}$, Pyk2 expression effect, $F_{(6,312)} = 20.06$, $P < 10^{-4}$. (c) Spine density and length were studied in similar conditions as in a, in the absence of treatment, using GFP or Pyk2:GFP fluorescence. (d) Quantification of spine density. One-way ANOVA: $F_{(6,155)} = 24.90$, $P < 10^{-4}$. (e) Quantification of spine length. One-way ANOVA: $F_{(6,157)} = 30.68$, $P < 10^{-4}$ and. In b,d,e, individual data points and means + s.e.m. are shown, 15–20 dendrites per condition (one–two dendrites per neuron) from two to three independent experiments. Post hoc multiple comparisons were done with Holm-Sidak's test (b,d,e), ***$P < 0.001$, ****$P < 10^{-4}$. Scale bars, 3 μm (a) and 1 μm (c).

GFP:Pyk2 rescued spine density in KO neurons *in vitro* (Fig. 6c,d). Transfection of GFP:Pyk2$_{1-840}$ had no significant effect, but, in contrast to what we observed for PSD-95 puncta rescue (see Fig. 6b), both GFP:Pyk2$_{Y402F}$ and GFP:Pyk2-KD fully restored spine density (Fig. 6c,d), revealing a role for Pyk2 independent of its autophosphorylation and kinase activity. We also quantified the effects of Pyk2 deletion on spine length (Fig. 6e). In the absence of Pyk2, spines were shorter, as observed *in vivo* (see Fig. 3i), but this effect was not rescued by re-expression of wild type or mutated Pyk2 (Fig. 6e). This lack of rescue of spine length deficits *in vitro* may indicate a contribution of presynaptic Pyk2 in spine length regulation since with the low

transfection rate in our culture system, concomitant transfection of pre- and post-synaptic neurons was very rare. Taken together, these results show that Pyk2 is important for PSD-95 synaptic enrichment and that this function requires both the C-terminal region involved in PSD-95 interaction and the autophosphorylation site and tyrosine kinase activity. In contrast, Tyr402 or kinase activity is not necessary for Pyk2 effects on spine density, revealing the existence of autophosphorylation/kinase activity-dependent and -independent roles of Pyk2 in spines.

**Hippocampal Pyk2 is altered in HD.** Since our results emphasized the high sensitivity of hippocampal function to Pyk2 protein

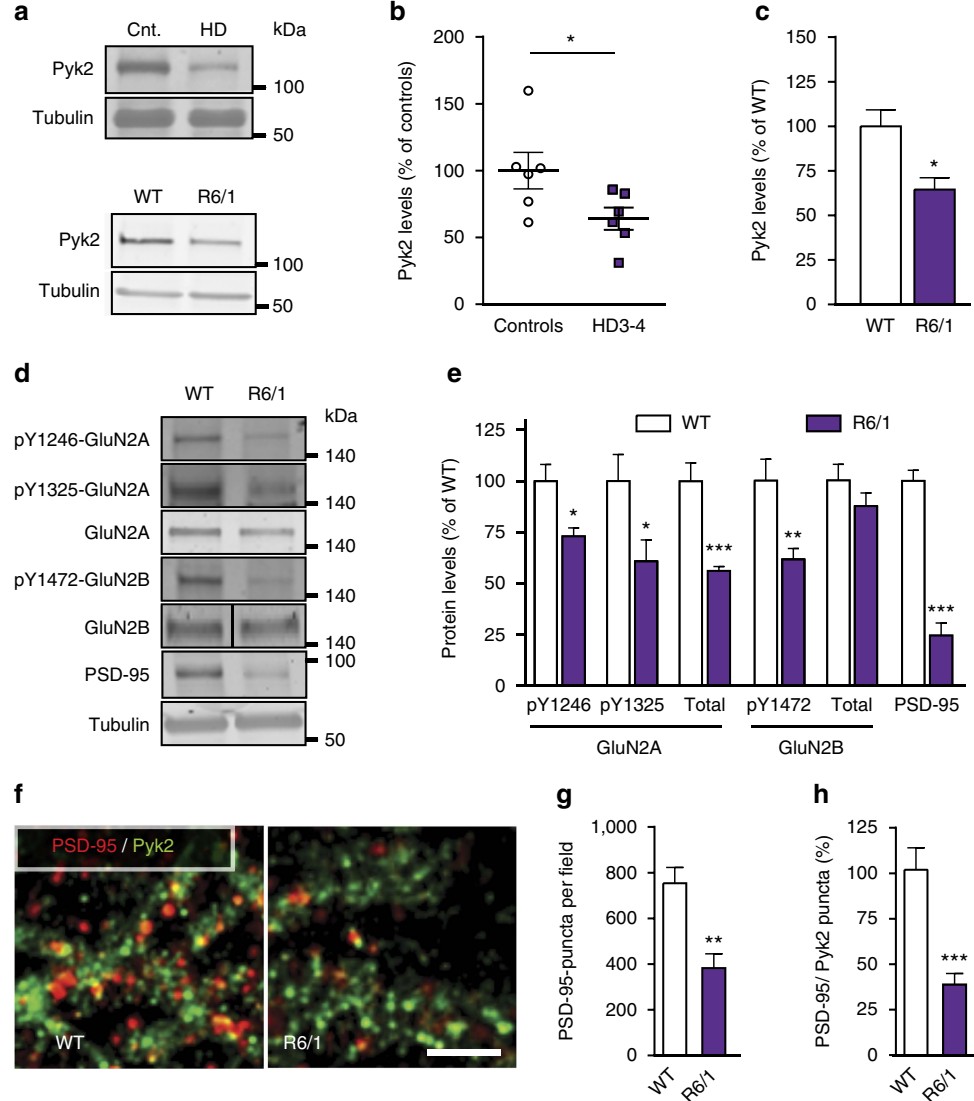

**Figure 7 | Hippocampal alterations of Pyk2 and synaptic markers in Huntington's disease.** (**a**) Hippocampal post-mortem samples from human patients grades 3–4 (HD3-4) and controls (Cnt., top panel) and from wild-type (WT) mice and R6/1 transgenic mice (lower panel) were analysed by immunoblotting for Pyk2 and α-tubulin as a loading control. Molecular weight marker positions are indicated in kDa. (**b**) Densitometric quantification of results as in **a**, for human samples expressed as a percentage of the mean in controls ($n = 6$ per group, Student's $t$-test, $t_{10} = 2.25$, $P < 0.05$). (**c**) Quantification of results as in **a** for WT and R6/1 mice (percentage of WT mean, $n = 4$–6 mice per group, Student's $t$-test, $t_8 = 3.23$, $P = 0.012$). (**d**) Immunoblotting for phosphorylated forms and total GluN2A and GluN2B, and PSD-95 in hippocampus of WT and R6/1 mice. (**e**) Quantification of results as in **d** (percentage of WT mean), Student's $t$-test, pY1246-GluN2A, $t_9 = 3.10$, $P = 0.013$, pY1325-GluN2A, $t_9 = 2.37$, $P = 0.04$, GluN2A, $t_9 = 5.21$, $P = 0.0006$, pY1472-GluN2B, $t_8 = 3.64$, $P = 0.0066$, GluN2B, $t_8 = 1.22$, $P = 0.26$, PSD-95, $t_9 = 9.18$, $P < 10^{-4}$. (**f**) Confocal images of the *stratum radiatum* of CA1 hippocampal sections from WT and R6/1 mice immunolabelled for PSD95 (red) and Pyk2 (green). Scale bar, 10 μm. (**g,h**) Quantification of results as in **f** (three slices per mouse, five–six mice per genotype. (**g**) Number of PSD95-positive puncta, Student's $t$-test, $t_9 = 3.98$, $P = 0.003$. (**h**) Number of Pyk2/PSD-95-double-positive puncta, expressed as a percentage of WT mean, Student's $t$-test, $t_{10} = 4.66$, $P = 0.0009$. All data are means + s.e.m. *$P < 0.05$, **$P < 0.01$ and ***$P < 0.001$. R6/1 mice were 5-month old. Uncropped blots for **a** and **d** are shown in Supplementary Figs 9 and 10, respectively.

expression levels, we hypothesized that any alteration in Pyk2 levels in pathological conditions might have deleterious consequences. HD appeared as an interesting condition since Pyk2 and wild-type Htt interact with the same SH3 domain of PSD-95 (refs 14,31). This interaction is altered in mutant Htt with a pathological polyglutamine expansion[31], resulting in PSD-95 mislocalization to extrasynaptic sites[32]. We noticed that the hippocampal phenotype of Pyk2 KO mice resembled that of HD mouse models, which display spatial learning impairments[33], decreased PSD-95 (ref. 34), dendritic spines loss[35] and shorter dendritic spine necks[33]. To test the possible involvement of Pyk2 in HD, we first measured Pyk2 protein levels in post-mortem

hippocampal samples from human patients. In patients with intermediate or late HD (grades 3–4 (ref. 36)) Pyk2 levels were reduced to $64 \pm 8\%$ of controls (mean ± s.e.m., Fig. 7a,b), whereas in patients at prodromal or early stage (grades 1–2) there was no significant change (Supplementary Fig. 4a,b). Pyk2 was also diminished in the hippocampus of R6/1 mice, an HD mouse model, transgenic for the first exon of the human *Htt* gene with amplified CAG repeats[37] ($64 \pm 6\%$ of control levels, mean ± s.e.m., Fig. 7a,c). Since Pyk2 can traffic between cytoplasm and nucleus[38], and since R6/1-mutated Htt accumulates in the nucleus[39] we looked for the existence of altered cytonuclear distribution of Pyk2 or its possible

sequestration in nuclear aggregates. Although Pyk2 was predominantly decreased in the cytoplasm (Supplementary Fig. 4c,d) it was not sequestered in intra-nuclear aggregates and did not co-localize with EM48-immunolabelled nuclear aggregates (Supplementary Fig. 4e). These results indicated a reduced Pyk2 function in the cytoplasm of HD mice. Indeed,

in R6/1 mice we observed changes similar to those in Pyk2 mutant mice, including a decrease in total GluN2A but no change in GluN2B, a decrease in tyrosine phosphorylated GluN2A and GluN2B, and a marked decrease in total PSD-95 (Fig. 7d,e). Double immunostaining for Pyk2 and PSD-95 in CA1 of wild-type and R6/1 mice showed a decreased number of

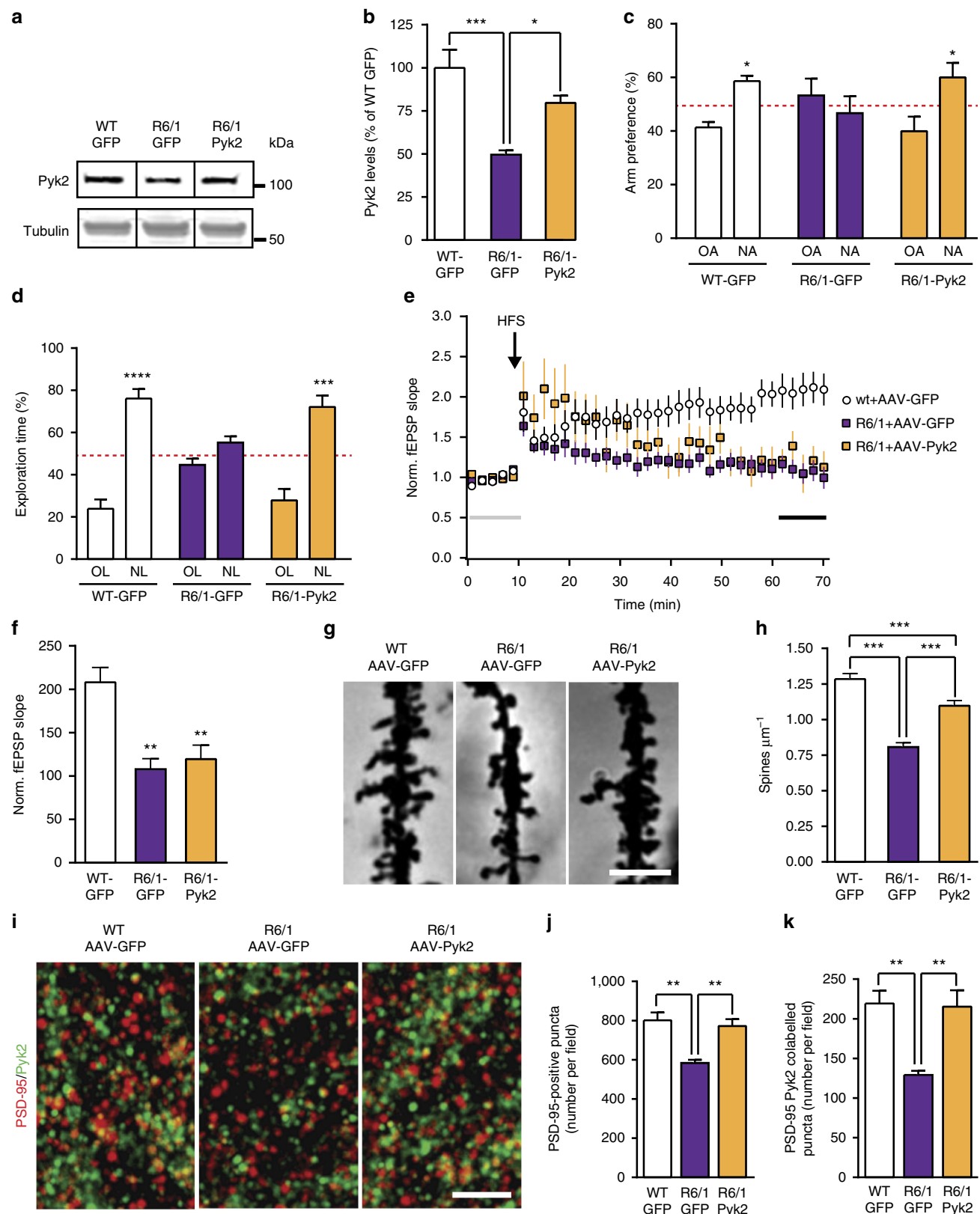

PSD-95-positive puncta in R6/1 mice as compared to wild-type mice (Fig. 7f,g), and less co-localization of PSD-95-positive and Pyk2-positive puncta in mutant mice (Fig. 7h). The similarity in the modifications observed in R6/1 mice and Pyk2 mutant mice suggested that the decreased levels of Pyk2 might contribute to PSD-95 and NMDA receptors subunits alterations.

**Pyk2 partly rescues the hippocampal phenotype of R6/1 mice.** Since the levels of Pyk2 in the hippocampus of HD patients and R6/1 mice (Fig. 7a–c) were close to those in Pyk2$^{+/-}$ mice, which displayed a similar phenotype, we asked whether correcting this defect in R6/1 mice could rescue some of their deficits. We stereotaxically injected AAV expressing either Pyk2 and GFP or GFP alone in the dorsal hippocampus of R6/1 mice (R6/1-Pyk2 and R6/1-GFP mice, respectively) or GFP alone in wild-type mice (WT-GFP) as a control. Three weeks after the injection, GFP expression demonstrated a wide spreading of viral transduction within the dorsal hippocampus, but restricted to this brain structure (Supplementary Fig. 4f). Immunoblotting showed that Pyk2 levels in R6/1-GFP mice, which were $50 \pm 3\%$ of those in WT-GFP mice, were raised to $80 \pm 4\%$ in R6/1-Pyk2 mice (Fig. 8a,b). The recovery of Pyk2 expression restored R6/1 mice performance in the Y-maze spontaneous alternation task (Fig. 8c) and corrected the deficit in the novel object recognition task (Fig. 8d). We also examined LTP in these mice at the same age as for behavioural experiments (4–5 months). In WT-GFP mice, we observed a robust LTP in CA1 after stimulation of Schaffer collaterals (Fig. 8e,f). In contrast, in R6/1-GFP and R6/1-Pyk2 mice, synaptic potentiation was not stable (Fig. 8e,f). One hour after high frequency stimulation (HFS), potentiation was observed only in WT-GFP mice (Fig. 8f), revealing that restoration of Pyk2 levels was not sufficient to correct the LTP impairment.

To assess the cellular effects of Pyk2 recovery possibly underlying the behavioural improvements, we analysed spine density in CA1 apical dendrites by the Golgi-Cox method (Fig. 8g). Spine density, which was decreased in R6/1-GFP compared to WT-GFP mice, was partly restored by Pyk2 viral expression in R6/1-Pyk2 mice (Fig. 8g,h). We also analysed PSD-95- and Pyk2-positive puncta in CA1 *stratum radiatum* of the three groups of mice (Fig. 8i). R6/1-GFP mice displayed a reduced number of PSD-95-positive puncta (Fig. 8j) and fewer double-positive PSD-95/Pyk2 puncta (Fig. 8k) compared to WT-GFP mice, consistent with the results shown in Fig. 7g–i. The numbers of PSD-95-positive puncta (Fig. 8j) and PSD-95/Pyk2 double-positive puncta (Fig. 8k) were both completely rescued in R6/1-Pyk2, reaching values similar to those in WT-GFP mice. These results strongly indicate that Pyk2 deficit contributes to the hippocampal phenotype of the R6/1 HD

mouse model, including cognitive deficits, dendritic spine loss and PSD-95 alteration. Moreover, they show that these deficits can be improved by Pyk2-induced expression, although this expression was not sufficient to restore synaptic plasticity in our experimental conditions.

## Discussion

Here we show the functional importance of the Ca$^{2+}$-activated non-receptor tyrosine kinase Pyk2 for hippocampal function and spines physiology. We also provide evidence that Pyk2 deficit contributes to the hippocampal impairments in a mouse model of HD, a severe genetic neurodegenerative disorder. Although Pyk2$^{+/-}$ and Pyk2$^{-/-}$ mice develop and breed normally in standard animal facility conditions, and show no gross behavioural defects, they appear strongly deficient in hippocampal-related memory tasks. These behavioural impairments were accompanied by impaired synaptic plasticity, decreased levels and/or tyrosine phosphorylation of NMDA receptor subunits, and alterations in PSDs composition and in spines density and morphology.

A previous study in hippocampal slices using overexpression or interfering constructs, reported that Pyk2 can regulate NMDA receptor function and LTP induction[7]. Here we show that in Pyk2 mutant mice LTP was not induced in standard conditions at Schaffer collaterals synapses on CA1 pyramidal neurons. A number of biochemical alterations at the post-synaptic level are likely to participate in this deficit. Both GluN2A and GluN2B were altered with a decreased total and tyrosine phosphorylated GluN2A and a decreased tyrosine phosphorylation of GluN2B. Moreover, subcellular fractionation revealed a decrease in the three NMDA receptor subunits (GluN1, GluN2A and GluN2B) in the PSD fraction. The reduction of GluN2B in PSDs may result from its decreased tyrosine phosphorylation, which is known to promote surface expression of GluN2B-containing NMDA receptors and their recruitment to PSDs[40,41]. This phosphorylation deficit was in agreement with the decreased active form of SFKs we observed in Pyk2 mutant mice, supporting their role in mediating NMDA receptors phosphorylation downstream from Pyk2 (ref. 7). In the case of GluN2A, the total protein and the phosphorylated form were decreased. Since tyrosine phosphorylation of GluN2A increases NMDA receptor currents[42,43], reduction of both forms of GluN2A may contribute to synaptic defects of Pyk2 knockout mice. These alterations seemed to be specific for the NMDA receptor complex since AMPA receptors levels and phosphorylation were not affected. These combined alterations in NMDA receptor subunits provide a first basis for the functional deficit in LTP induction. Many other aspects of synaptic function and plasticity remain to be investigated in Pyk2 mutant mice, and

**Figure 8 | Pyk2 protein levels restoration in the hippocampus partly rescues R6/1 mice phenotype.** (**a**) Pyk2 and α-tubulin (loading control) immunoblotting in 3-month WT mice injected with AAV-GFP (WT-GFP), or R6/A injected with AAV-GFP (R6/1-GFP) or AAV-Pyk2 and GFP (R6/1-Pyk2). Uncropped blots in Supplementary Fig. 11. (**b**) Quantification of results as in **a** (six–nine mice per group). One-way ANOVA: $F_{(2,18)} = 4.39$, $P < 0.05$, Holm-Sidak's test versus R6/1-GFP. (**c**) Y-maze spontaneous alternation test (10–11 mice per group). Two-way ANOVA interaction $F_{(2,56)} = 4.39$, $P < 0.05$, OA versus NA, Holm-Sidak's test WT-GFP $t = 2.64$, $P < 0.05$, R6/1-GFP, $t = 0.97$, ns, R6/1-Pyk2, $t = 2.93$, $P < 0.05$. (**d**) NOL test (9–12 mice per group). Two-way ANOVA interaction $F_{(2,54)} = 11.9$, $P < 0.0001$, OL versus NL, Holm-Sidak's test WT-GFP $t = 9.08$, $P < 0.0001$, R6/1-GFP, $t = 1.60$, ns, R6/1-Pyk2, $t = 6.66$, $P < 0.0001$. (**e**) LTP studied as in Fig. 1c in hippocampal slices from 5-month WT-GFP, R6/1-GFP and R6/1-Pyk2 mice ($n = 3$–4 mice per group, 2–3 slices per mouse, 10–11 slices total). (**f**) Ten-min average of fEPSP slope 40 min after HFS, normalized to the mean of 10-min baseline (corresponding time points are indicated in **e** by an horizontal line). Kruskal–Wallis $= 15.63$, $P < 0.05$, *post hoc* analysis with Dunn's multiple comparisons test. (**g**) Golgi-Cox-stained apical dendrites from CA1 pyramidal neurons. Scale bar, 3 μm. (**h**) Quantitative analysis of dendritic spine density as in **e** (59–62 dendrites from four mice per group). One-way ANOVA: $F_{(2,177)} = 46.7$, $P < 10^{-4}$, Holm-Sidak's test versus R6/1-GFP. (**i**) Hippocampal sections of WT and R6/1 mice injected with AAV-GFP or AAV-Pyk2 as indicated, and double-immunostained for PSD-95 and Pyk2. High magnification in CA1 *stratum radiatum* is shown. Scale bar, 5 μm. (**j**) Quantification of PSD-95-positive puncta density. One-way ANOVA $F_{(2,14)} = 10.81$, $P = 0.0014$, Holm-Sidak's multiple comparisons test. (**k**) Quantification of PSD-95/Pyk2 double-positive puncta density. One-way ANOVA $F_{(2,14)} = 9.76$, $P = 0.0022$, Holm-Sidak's multiple comparisons test. In **j,k**, five–seven mice per group. In all graphs values are means + s.e.m. *$P < 0.05$, **$P < 0.01$, ***$P < 0.001$ and ****$P < 10^{-4}$.

the modifications induced by the absence of Pyk2 clearly extend beyond NMDA receptors.

A marked alteration observed in Pyk2 mutant mice concerned PSD-95. PSD-95 SH3 domain is known to bind Pyk2 C-terminal Pro-rich motif[14], thereby clustering and activating Pyk2 in response to $Ca^{2+}$ increase[5,14]. In contrast, effects of Pyk2 on PSD-95 have not been described. Our study reveals that Pyk2 has a critical influence on PSD-95, regulating its levels, its localization at PSDs in basal conditions and its clustering in response to stimulation of NMDA receptors. The decreased PSD-95 expression cannot explain, by itself, the absence of LTP in Pyk2 mutant mice, since PSD-95 knockout mice display an enhanced LTP[44]. Therefore, the functional deficit is likely to result from its combination with dysregulation of NMDA receptors and possibly other proteins. PSD-95 is phosphorylated by c-Abl and SFKs on several tyrosine residues, which can favour PSD-95 aggregation and GluN2A activation[28,45]. Thus, there appears to be a reciprocal interaction between Pyk2 and PSD-95, each enhancing the function of the other, thereby directly and indirectly regulating NMDA receptors and PSD organization. In support of this functional association, it has been reported that in hippocampal neurons in culture, corticosterone-induced recruitment of Pyk2, PSD-95 and GluN1 to spines requires Pyk2 activation[46]. NMDA receptor activation recruits Pyk2 to spines through its interaction with PSD-95 (ref. 5), whereas it also rapidly destabilizes PSD-95 and removes it from PSDs[47]. Our study shows that Pyk2 is required for the later recruitment of PSD-95 to spines revealing its contribution in the coordinated $Ca^{2+}$-dependent dynamics of PSD proteins, a key aspect of synaptic function and plasticity. Importantly, GluN2A, GluN2B and PSD-95 co-assemble together with PSD-93 into 1.5 MDa 'supercomplexes'[48]. PSD-93 is phosphorylated by Fyn[49], which is altered in PyK2 mutant mice. Since both PSD-93 and PSD-95 promote Fyn-mediated tyrosine phosphorylation of GluN2A and/or GluN2B[50,51], it appears that Pyk2 is potentially placed at a strategic location to regulate NMDA receptor supercomplexes. Since these supercomplexes have been proposed to be functionally important, with mutations in their key components resulting in abnormal LTP and learning[48], it will be particularly interesting to investigate whether their alteration accounts for LTP and other synaptic deficits in Pyk2 mutant mice.

Dendritic spine density and length were also altered in Pyk2 mutant mice. Our study of the requirement of Pyk2 protein domains and functions for its various roles reveals the complexity of its contribution. The kinase activity of Pyk2 and its autophosphorylation site, Tyr402, which is critical for the recruitment of SFKs, were both necessary for the rescue of PSD-95 clustering in spines in Pyk2 KO neurons. The C-terminal region, which allows interaction with PSD-95 (ref. 14), was also required for PSD-95 clustering. In contrast, rescue of spine density required the C-terminal region but neither Tyr402 nor the kinase activity, indicating a phosphorylation-independent role of Pyk2 in the regulation of spine number. Such an autophosphorylation/kinase activity-independent function of Pyk2 is reminiscent of those reported for the closely related FAK[52,53] in non-neuronal cells. This alternate signalling is presumably SFK-independent and may be linked to scaffolding properties of Pyk2 and/or its interaction with specific partners.

The current study also suggests a role for Pyk2 at the presynaptic level. Our electron microscopy experiments show the presence of Pyk2 in nerve terminals, confirming previous biochemical observations[54]. Its functional role is indicated by the alteration in paired-pulse facilitation, a form of short-term plasticity that is considered to be mostly presynaptic[55,56], although post-synaptic mechanisms are also possible[57]. Finally, in the absence of Pyk2, spine length was decreased both *in vivo*

and in culture. A possible explanation for this lack of rescue could be a dual role of Pyk2 at the pre- and post-synaptic levels, which were not simultaneously restored at the same synapses in the culture conditions in which the transfection rate was low. At any rate, the mechanisms by which Pyk2 controls spine length are likely to be complex since both positive and negative effects of Pyk2 and FAK on spine growth have been previously reported[58–60] and their elucidation will require further investigation. The important aspect of the present results is the overall deficit in spine number and length in the absence of Pyk2. Interestingly, it has been shown that chronic stress induces a redistribution of activated Pyk2 to the perinuclear region of CA3 neurons, contributing to a deficit in the nuclear pore protein NUP62 and its potential negative consequences on dendritic complexity[61]. The present study suggests that Pyk2 redistribution could also directly contribute to dendritic or synaptic alterations by reducing its local levels.

Our observations thus reveal an essential and complex role of Pyk2 in the regulation of spines, PSDs and NMDA receptors, whose alterations impair synaptic plasticity and hippocampal-dependent memory in Pyk2-deficient mice. The importance of Pyk2 expression levels is underlined by the unexpected severity of the functional deficits in heterozygous mutant mice. Their behavioural and physiological phenotype was as severe as in homozygous mutant mice, whereas a clear gene dosage effect was observed at the molecular level. This suggests that Pyk2 expression levels are a limiting factor for excitatory synapses function in hippocampus and raises the question of the possible implications of decreased Pyk2 protein levels in pathological conditions. This hypothesis was supported by our results in a mouse model of HD. Pyk2 was decreased in the R6/1 mouse, to a level comparable to that observed in Pyk2$^{+/-}$ heterozygous mutant mice, which displayed a clear behavioural phenotype. Several alterations in R6/1 mice were similar to those in Pyk2 mutant mice including the alterations in NMDA receptors, PSD-95 distribution and spines. Of course, such alterations could potentially result from different mechanisms in the two types of mutant mice, but our results provide strong evidence that Pyk2 deficit contributes to some of the abnormalities observed in R6/1 mice. Although enhancing Pyk2 expression by AAV transduction in R6/1 mice was not sufficient to restore a normal LTP in CA1, it corrected several behavioural, molecular and cellular deficits. Interestingly, the deleterious consequences of Pyk2 deficit are likely to synergize with other factors including increased activity of STEP[62], a tyrosine phosphatase active on Pyk2 (ref. 63), which is expected to aggravate the functional consequences of Pyk2 insufficiency. In patients, the role of Pyk2 deficiency remains to be determined since it was detected in grades 3–4 when there is a loss of other synaptic proteins but not at earlier stages (grades 1–2). Nevertheless, our results suggest that strategies for enhancing Pyk2 expression or activity, or for inhibiting STEP phosphatase activity[64] could have a potential therapeutic interest in HD. Further work will determine whether Pyk2 deficiency could play a role in other neurodegenerative conditions, such as Alzheimer disease[8], besides its possible role as modulator of Tau toxicity[65].

Our study reveals that the absence of Pyk2 impairs synaptic functions and hippocampal-dependent learning and memory. We show that Pyk2 plays critical roles in spines and PSD organization and in the regulation of PSD-95 and NMDA receptors. Although we focused our investigations on hippocampus where Pyk2 expression is the highest, it is likely that it is also important in other neurons, especially in neocortical areas where it is highly expressed and which are known to undergo intense synaptic plasticity. We also reveal the contribution of Pyk2 in hippocampal dysfunction in a HD model and its potential

reversibility. Our results should stimulate research on the role of Pyk2 in other pathological conditions in which NMDA receptor dysfunction is thought to be directly or indirectly involved.

## Methods

**Animals.** For Pyk2 deletion, Pyk2$^{f/f}$ C57Bl/6 mice were generated in which the *PTK2B* exons 15b-18 were flanked with LoxP sequences (Gen-O-way, Lyon, France) leading to a deletion that disrupts the protein kinase domain when crossed with a expressing Cre line[20]. Pyk2$^{-/-}$ mice and Pyk2$^{f/f}$ mice were genotyped from a tail biopsy (Charles River, Saint-Germain-Nuelles, France) using for DNA amplification of floxed *PTK2B* as forward primer 5′-GAGAGTGCTGGGT ACTCCAGACTCAGATAG-3′ and as reverse primer 5′-TTCAGGAACACCAG AGAACTAGGGTGG-3′ and previously reported primers[20] for Pyk2$^{-/-}$ mice. Heterozygous mice were crossed to generate $+/+$, $+/-$ and $-/-$ mice. Male R6/1 transgenic mice[37] (4–5-month-old) expressing exon-1 mutant Htt with 145 glutamines under the HD human promoter and their wild-type littermates were obtained from Jackson Laboratory (Bar Harbor, ME, USA). Housing room was kept at 19–22 °C and 40–60% humidity, under a 12:12 h light/dark cycle and mice had ad libitum access to food and water. Animal experiments and handling was in accordance with ethical guidelines of Declaration of Helsinki and NIH, (1985-revised publication no. 85-23, European Community Guidelines), and French Agriculture and Forestry Ministry guidelines for handling animals (decree 87849, licence A 75-05-22) and approval of the Charles Darwin ethical committee. All mice used in this study were males and the ages are indicated in the figure legends.

**Behavioural phenotyping.** Wire hanging, plus maze and open-field paradigms were carried out as described elsewhere[66]. In brief, for wire hanging, mice were placed on a standard wire cage lid, which was then turned upside down for 60 s the number of falls was recorded. The elevated plus maze consisted of two opposing 30 × 8 cm open arms, and two opposing 30 × 8 cm arms enclosed by 15 cm high walls. The maze was raised 50 cm from the floor and lit with dim light. Each mouse was placed in the central square, facing an open arm and the time spent in the open arms was recorded for 5 min. The open field apparatus was a dimly lit white circular arena (40 cm diameter, 40 cm high wall). Mice were placed in the arena centre and free exploration was recorded for 10 min. NOL test and spontaneous alternation in a Y-maze task (Y-SAT) were performed as previously described[33]. Briefly, for NOL an open-top arena (45 × 45 × 45 cm) was used. Mice were first habituated to the arena (2 days, 15 min per day). On day 3 of the acquisition phase, two identical objects (A1 and A2) were placed in the arena and explored for 10 min. Twenty-four hour later, one object was moved from its original location to the diagonally opposite corner and mice were allowed to explore the arena for 5 min. The object preference was measured as the time exploring each object × 100/time exploring both objects. For Y-SAT, a Y-maze apparatus, made of clear Perspex, was used (Y-maze dimensions: arms, 35 cm length, 25 cm height, 15 cm width). For the training session, mice were placed in the stem arm of the Y and allowed to explore for 10 min only one accessible arm (familiar arm) for 10 min. They were then returned to their home cage. After an inter-trial interval of 2 h, mice were placed in the stem arm of the T-maze and allowed to freely explore all three arms for 5 min. Big and highly perceptible objects were situated surrounding the maze at 20–40 cm. The arm preference was measured as the time exploring each arm × 100/time exploring both arms. Animals were tracked and recorded with Smart junior software (Panlab).

**Viral constructs and stereotaxic injection.** For specific deletion of Pyk2 in dorsal hippocampus expression, 4-week-old Pyk2$^{f/f}$ mice were stereotaxically injected with AAV expressing Cre recombinase and (AV-9-PV2521, AAV9.CamKII.HI.eGFP-Cre.WPRE.SV40 (AAV-Cre)) (from Perelman School of Medicine, University of Pennsylvania, USA). To overexpress Pyk2, we used AAV1-CamKIIa0.4-GFP-2A-mPTK2B (AAV-Pyk2) with a T2A cleavable link (Vector Biolabs Malvern, PA, USA). As a control, we injected AAVs expressing GFP (AV-9-PV1917, AAV9.CamKII0.4.eGFP.WPRE.rBG (AAV-GFP) from Perelman). Following anaesthesia with pentobarbital (30 mg kg$^{-1}$), we performed bilateral injections of AAV-GFP, AAV-Cre or AAV-Pyk2 (2.6 × 10$^9$ GS per injection) in the dorsal hippocampus following coordinates from the bregma (millimetres); anteroposterior, −2.0; lateral, ± 1.5; and dorsoventral, ± 0.8. For R6/1 mice, we performed an additional injection at dorsoventral ± 1.2. AAV injection was carried out in 2 min. The cannula was left in place for 5 min for complete virus diffusion before being slowly pulled out of the tissue. After 2 h of careful monitoring, mice were returned to their home cage for 3 weeks before starting analyses of behaviour, biochemistry and morphology.

**Electrophysiology.** Littermate mice (1–2 months) received an injection of ketamine/xylazine (75/10 mg kg$^{-1}$, i.p.) and intracardially perfused with an ice-cold solution ((all mM) 25 NaHCO$_3$, 1.25 NaH$_2$PO$_4$, 2.5 KCl, 0.5 CaCl$_2$, 7 MgCl$_2$, 110 choline chloride, 25 glucose, 11.6 ascorbic acid, and 3.1 pyruvic acid). The brain was rapidly removed and horizontal slices (350 μm thick) cut with a vibratome (Microm, Thermo Fisher). Slices were placed in an interface chamber

containing artificial CSF (ACSF) ((all mM) 124 NaCl, 1 NaH$_2$PO$_4$, 26.2 NaHCO$_3$, 2.5 KCl, 1.6 CaCl$_2$, 1.2 MgCl$_2$, 11 glucose) at 37 °C, saturated with 5% (vol/vol) CO$_2$ in O$_2$, and allowed to recover for 1 h. Slices were then transferred to a submerged recording chamber, a cut was made between the CA3 and CA1, and bicuculline-supplemented ACSF used for superfusion. An ACSF-filled recording borosilicate glass pipette (2–4 MΩ) was inserted in the *stratum radiatum* of CA1 region. Schaffer collaterals were stimulated (HFS, 5 × 1 s at 100 Hz) with a tungsten bipolar electrode (0.5 MΩ). Field excitatory post-synaptic potentials (fEPSPs) were recorded using a multiclamp 700B amplifier (Molecular Devices) low-pass filtered at 5 kHz and digitized at 20 kHz. Offline analysis of fEPSP slopes was done with Clampfit software (Molecular Devices). Baseline potential was set to zero and recordings were low-pass filtered at 1 kHz with Bessel filter. A 1-ms time-window was manually positioned at the onset of the fEPSP and its initial slope automatically measured. LTP was investigated in AAV-injected R6/1 mice in similar conditions except that mice were 5-month old at the time of the recording. Paired pulse ratio was determined at a 50 ms interval. In all cases, the experimenter carrying out the acquisition and analysis of electrophysiological data was blind to the mice genotype and/or AAV type.

**Electronic microscopy.** Mice were transcardially perfused with a solution containing 40 g l$^{-1}$ paraformaldehyde and 1 ml l$^{-1}$ glutaraldehyde in 0.1 M sodium phosphate buffer (PB), pH 7.4. Brains were then immersed in the same fixative 12 h at 4 °C. Tissue blocks containing the hippocampus were dissected and washed in 0.1 M PB, cryoprotected in 10 and 20% sucrose in 0.1 M PB, freeze thawed in isopentane and liquid nitrogen. Samples were post fixed in 25 ml l$^{-1}$ glutaraldehyde in 0.1 M PB for 20 min, washed and treated with 20 g l$^{-1}$ osmium tetroxide in PB for 20 min. They were dehydrated in a series of ethanol and flat embedded in epoxy resin (EPON 812 Polysciences). After polymerization, blocks from the CA1 region were cut at 70 nm thickness using an ultramicrotome (Ultracut E Leica). Sections were cut with a diamond knife, picked up on formvar-coated 200 mesh nickel grids. For etching resin and remove osmium, sections were treated with saturated aqueous sodium periodate (NaIO$_4$). They were then immunostained for Pyk2 with rabbit antibodies (see below) by indirect immunolabelling protein A-gold probes (20 nm) (CMC Utrecht; Netherlands) following a published method[67]. The sections were then double stained with uranyl acetate and lead citrate before observation with a Philips (CM-100) electron microscope. Digital images were obtained with a CCD camera (Gatan Orius). To test the immunostaining specificity, the primary antibody was omitted.

**Tissue preparation and immunofluorescence.** Mice were anaesthetized (pentobarbital, 60 mg kg$^{-1}$ i.p.) and intracardially perfused with a 40 g l$^{-1}$ paraformaldehyde solution in 0.1 M sodium phosphate, pH 7.2. Brains were removed and post-fixed overnight in the same paraformaldehyde solution, cryoprotected with 300 g l$^{-1}$ sucrose in PBS with 0.2 g l$^{-1}$ NaN$_3$ and frozen in dry-ice cooled isopentane. All the following steps were done with gentle shaking. Serial 30-μm coronal cryostat-free floating sections were washed three times in PBS, permeabilized in PBS with 3 ml l$^{-1}$ Triton X-100 and 30 ml l$^{-1}$ normal goat serum (Pierce Biotechnology, Rockford, IL, USA) for 15 min at room temperature, and washed three times. Brain slices were then incubated overnight at 4 °C in the presence of primary antibodies in PBS with 0.2 g l$^{-1}$ NaN$_3$: rabbit Pyk2 antibody (1:500, #07M4755) and mouse MAP2 antibody (1:500) from Sigma, Chemical Co. (St Louis, MO, USA), mouse antibodies for Htt (clone EM48 1:150, #2026373, Chemicon, Temecula, CA, USA), mouse anti-PSD-95 1:500 (#QA210648, Thermo Scientific, MA, USA). Sections were then washed three times and incubated for 2 h at room temperature with fluorescent secondary antibodies: Cy3 goat anti-rabbit (1:200) and/or AlexaFluor 488 goat anti-mouse (1:200; both from Jackson ImmunoResearch, West Grove, PA, USA). No signal was detected in control sections incubated in the absence of the primary antibody.

**Primary hippocampal neurons culture and immunofluorescence.** Hippocampal neurons were prepared from E17 C57Bl/6 mouse embryos (pregnant mice from Charles River, Saint Germain Nuelles, France) or from our Pyk2 mice colony as previously described[33]. The neuronal cell suspension was seeded (70,000 cells cm$^{-2}$) on coverslips precoated with poly-D-lysine (0.1 mg ml$^{-1}$, Sigma) in 24-well plates or in 6-well plates without coverslips. Neurobasal medium (GIBCO, Renfrewshire, Scotland, UK) containing 1 ml per 50 ml of B27 supplement (Gibco-BRL) and 50 ml of GlutaMAX (100 ×) (Gibco-BRL) was used to grow the cells in serum-free medium conditions and maintained at 37 °C in 5% CO$_2$. At DIV 21–22, cells were treated with vehicle or 10 μM MK801 (Sigma) for 30 min. Then, cells were treated with vehicle or 40 μM glutamate (Sigma) for 15 min and samples were collected for immunoblot analysis or the glutamate was washed out and cells further incubated for 3 h before being fixed for 10 min with 40 g l$^{-1}$ paraformaldehyde in PB 0.2 M for immunostaining. Fixed cells were permeabilized in 1 ml l$^{-1}$ Triton X-100 for 10 min and then blocking was performed with 10 g l$^{-1}$ BSA in PBS for 1 h. Cells were incubated with mouse monoclonal antibodies for PSD-95, (1:500, #QA210648, Millipore) or MAP2 (1:800, #073M4774, Sigma) or rabbit Pyk2 antibodies (1:500, #074M4755, Sigma, XX) at 4 °C overnight. After three washes with PBS, cells were incubated with the

corresponding fluorescent secondary antibodies, Cy3 or Cy2 (1:200; Jackson ImmunoReseach, West Grove, PA). Then, cells were rinsed twice with PBS and incubated with phalloidin–rhodamine 1:1,000 (Sigma) for 45 min in PBS. After washing twice with PBS, the coverslips were mounted with Vectashield (Vector Laboratories Burlingam, UK). Hippocampal neuron staining was observed with a confocal SP5-II (see below).

**Cell transfection and constructs.** Pyk2$^{+/+}$ and Pyk2$^{-/-}$ hippocampal neurons at DIV 18 were transfected using transfectin (Bio-Rad, Hercules, CA, USA) following the manufacturer's instructions and left for 48–72 h. Cells were transfected with previously described constructs[38]: GFP (control), GFP-Pyk2, GFP-Pyk2$^{1-840}$ (Pyk2 deleted from the FAT domain and the third proline-rich motif) and GFP-Pyk2$^{Y402F}$ (Pyk2 with a point mutation of the autophosphorylated tyrosine-402). GFP was fused to the N terminus of Pyk2.

**Confocal imaging and analysis.** Immunostained neurons in culture or tissue sections were imaged using a Leica Confocal SP5-II (63 × numerical aperture lens, 5 × digital zoom, 1-Airy unit pinhole. Four frames were averaged per z-step throughout the study. Confocal z-stacks were taken at 1,024 × 1,024 pixel resolution, every 2 μm for brain sections and every 0.2 μm for cultured cells. The labelled PSD95-positive clusters number and size were quantified with NIH ImageJ freeware (Wayne Rasband, NIH) as described[68] with minor changes. For tissue sections, at least three 30-μm slices containing dorsal hippocampus were analysed per mouse and up to three representative CA1 *stratum radiatum* images were obtained from each slice. For cultured hippocampal neurons, PSD95-positive clusters co-localized with F-actin clusters (stained with phalloidin–rhodamine) or GFP-enriched spines were quantified as described[69] with minor changes using ImageJ. The number of neurites analysed was >20 neurites (one–two neurites per neuron) per condition from two–three different cultures.

**Golgi staining and spine analysis.** Fresh brain hemispheres were processed following the Golgi-Cox method as described elsewhere[70]. Essentially, mouse brain hemispheres were incubated in the dark for 14–17 days in filtered dye solution (10 g l$^{-1}$ K$_2$Cr$_2$O$_7$, 10 g l$^{-1}$ HgCl$_2$ and 8 g l$^{-1}$ K$_2$CrO$_4$). The tissue was then washed 3 × 2 min in water and 30 min in 90% EtOH (v/v). Two hundred-μm sections were cut in 70% EtOH on a vibratome (Leica) and washed in water for 5 min. Next, they were reduced in 16% ammonia solution for 1 h before washing in water for 2 min and fixation in 10 g l$^{-1}$ Na$_2$S$_2$O$_3$ for 7 min. After a 2- min final wash in water, sections were mounted on superfrost coverslips, dehydrated for 3 min in 50%, then 70, 80 and 100% EtOH, incubated for 2 × 5 min in a 2:1 isopropanol:EtOH mixture, followed by 1 × 5 min in pure isopropanol and 2 × 5 min in xylol. Bright-field images of Golgi-impregnated *stratum radiatum* dendrites from hippocampal CA1 pyramidal neurons were captured with a Nikon DXM 1200F digital camera attached to a Nikon Eclipse E600 light microscope (× 100 oil objective). Only fully impregnated pyramidal neurons with their soma entirely within the thickness of the section were used. Image z-stacks were taken every 0.2 μm, at 1,024 × 1,024 pixel resolution, yielding an image with pixel dimensions of 49.25 × 49.25 μm. Z-stacks were deconvolved using the Huygens software (Scientific volume imaging, Hilversum, the Netherlands) to improve voxel resolution and to reduce optical aberration along the z axis. Segments of proximal apical dendrites were selected for the analysis of spine density and spine morphology according to the following criteria: (a) segments with no overlap with other branches that would obscure visualization of spines and (b) segments either 'parallel' to or 'at acute angles' relative to the coronal surface of the section to avoid ambiguous identification of spines. Only spines arising from the lateral surfaces of the dendrites were included in the study; spines located on the top or bottom of the dendrite surface were ignored. Given that spine density increases as a function of the distance from the soma, reaching a plateau 45 μm away from the soma, we selected dendritic segments of basal dendrites 45 μm away from the cell body. The total number of spines was obtained using the cell counter tool in the ImageJ software. At least 60 dendrites per group from at least three mice per genotype were counted. For a more precise description of the dendritic shape changes, the spine head diameter was analysed as a continuous distribution (between 368 and 418 spines per group were analysed) using the ImageJ software. Then, a distribution analysis of head diameter was performed. Then, head diameter analysis was performed manually using ImageJ for all the spines in control mice. Spine neck was measured in all spines as the distance from the dendritic shaft to the head of the spine using the ImageJ.

**Subcellular fractionation.** Hippocampus from 20-week-old R6/1 mice was homogenized with a Teflon-glass potter in lysis buffer (4 mM HEPES, 0.32 M sucrose, 1 mM phenylmethylsulfonyl fluoride (PMSF), 10 mg l$^{-1}$ aprotinin, 1 mg l$^{-1}$ leupeptin, 2 mM sodium orthovanadate, 0.1 g l$^{-1}$ benzamidine) and centrifuged at 3,000 g for 10 min. The supernatant was taken as 'cytosolic' fraction and the pellet as 'nuclear' fractions. The latter was resuspended in 10 mM Tris-HCl (pH 7.5), 0.25 M sucrose, 2 mM PMSF, 10 mg l$^{-1}$ aprotinin, 1 mg l$^{-1}$ leupeptin, 2 mM Na$_3$VO$_4$ and sonicated. For PSD preparation, the two hippocampi (30–50 mg) of each mouse were homogenized using a Dounce Homogenizer in 1 ml of homogenization buffer (320 mM sucrose, 1 mM Hepes pH 7.4). Samples were

centrifuged at 1,000 x g for 10 min, 4 °C. The pellet was resuspended in 1 ml of homogenization buffer and centrifuged at 1,000 x g for 10 min, 41 °C. The two supernatants were pooled and centrifuged at 18,600 x g for 15 min, 4 °C. The pellet was then resuspended in 1 ml of 1.5 M sucrose, 50 mM Tris-HCl pH 7.4) and transferred in a 4 ml. Thinwall, ultra-clear Beckman Coulter tube and centrifuged on a 0.3–0.85 sucrose gradient at 25,000 r.p.m. for 85 min, 4 °C (SW60 Ti rotor in Optima L90K Ultracentrifuge). The synaptosomal fraction was collected at the interface, resuspended in 2 ml 50 mM Tris-HCl, pH 7.4 and centrifuged at 32,000 r.p.m. for 18 min, 4 °C (MLA-80 rotor, Optima Max Ultracentrifuge). The pellet was resuspended in 1 ml Tris-HCl, 25 mM pH 7.4, 30 mM l-1 Triton X100 and left on ice for 30 min. Then the sample was loaded onto 2 ml of Tris buffer with 0.85 M sucrose in a Thinwall, ultra-clear tube and centrifuged at 32,000 r.p.m. for 40 min, 4 °C (SW60 Ti rotor in Optima L90K Ultracentrifuge). The pellet corresponding to the PSD fraction was resuspended in 50 mM Tris-HCl, pH 7.4. Until the gradient buffers contained 30 mM NaF, 5 mM sodium-orthovanadate, and cOmpleteTM ULTRA protease inhibitor cocktail (Roche).

**Post-mortem human brain tissues.** Hippocampal brain tissues were supplied by the Banc de Teixits Neurològics (Biobanc-HC-IDIBAPS), Barcelona, Spain. They included six controls (mean ± s.e.m.; two females, four males, age 53.5 ± 6.8 years, post-mortem intervals, 4–18 h), four patients with HD grades 1–2 (four males, age 72.2 ± 1.7 years post-mortem intervals, 6–14 h) and seven patients with HD grades 3–4 (three females, four males, age 54.5 ± 6.5 years, post-mortem intervals of 4–17 h).

**Immunoblot analysis.** Animals were killed by cervical dislocation. The hippocampus was dissected out, frozen using CO$_2$ pellets and stored at − 80 °C until use. Briefly, the tissue was lysed by sonication in 250 μl of lysis buffer (PBS, 10 ml l$^{-1}$ Nonidet P-40, 1 mM PMSF, 10 mg l$^{-1}$ aprotinin, 1 mg l$^{-1}$ leupeptin and 2 mg l$^{-1}$ sodium orthovanadate). After lysis, samples were centrifuged at 12,000 r.p.m. for 20 min. Supernatant proteins (15 μg) from total brain regions extracts were loaded in SDS–PAGE and transferred to nitrocellulose membranes (GE Healthcare, LC, UK). Membranes were blocked in TBS-T (150 mM NaCl, 20 mM Tris-HCl, pH 7.5, 0.5 ml l$^{-1}$ Tween 20) with 50 g l$^{-1}$ phospho-Blocker (Cell Biolabs, San Diego, CA) or 50 g l$^{-1}$ non-fat dry milk and 5 g l$^{-1}$ BSA. Immunoblots were probed with the following antibodies (all diluted 1:1,000): rabbit polyclonal antibodies: Pyk2 (#074M4755, Sigma), Pyk2 (#ab32571, Abcam, epitope within the first 100 residues), phosphoY402-Pyk2 (#5), PSD-95 (#QA210648), phosphoY876-GluA2 (#2), and phosphoY1246-GluN2A (#1, Cell Signaling Technology, Danvers, MA, USA), GluA1 (#JBC1830522, Upstate Biotechnology, NY, USA), phosphoY1472-GluN2B (#04242010009761, Cayman antibodies, Ann Arbor, MI, USA), phosphoY418-Src reacting with all phosphoSFKs (#GR144140-2), and phosphoY1325-GluN2A (#GR14161032, Abcam, Cambridge, UK), phosphoS831-GluA1 (#2726818), GluN2B (#2697434), GluA2 (#2280905), and GluN2A (#NRG1815904, Millipore Bedford, MA, USA), mouse monoclonal antibodies: phosphoERK1/2 (#26,Cell Signaling Technology, Danvers, MA, USA), FAK (#JBC1900835, Santa Cruz Biotechnology, Santa Cruz, CA, USA), GluN1 (#225310,Millipore, Bedford, MA, USA). All blots were incubated with the primary antibody overnight at 4 °C by shaking in PBS with 0.2 g l$^{-1}$ sodium azide. After several washes in TBS-T, blots were incubated with secondary anti-rabbit or anti-mouse IgG IRdye800CW-coupled or anti-mouse IgG IRdye700DX-coupled antibodies (1:2,000, Rockland Immunochemicals, USA). Secondary antibody binding was detected by Odyssey infrared imaging apparatus (Li-Cor Inc., Lincoln, NE). For loading control a mouse monoclonal antibody for α-tubulin was used (#083M4847V, 1:10,000; Sigma).

**Statistical analysis.** Statistical analyses were carried out using the GraphPad Prism 6.0 software. Data sets were tested for normality distribution with d'Agostino–Pearson and Shapiro–Wilk tests. When distribution was not different from normal, they were analysed with parametric using Student's t-test (95% confidence), one-way ANOVA or two-way ANOVA, with Holm–Sidak *post hoc* multiple comparisons test. Two by two comparisons were two-tailed. In cases in which the distribution was significantly different from normal ($P < 0.05$), non-parametric tests were used including Mann and Whitney for two groups comparisons and Kruskal–Wallis for more than two groups and Dunn's test for *post hoc* multiple comparisons. Kolmogorov–Smirnov test was used as indicated in the figure legends. Values of $P < 0.05$ were considered as statistically significant.

**Data availability.** The authors declare that the data supporting the findings of this study are available within the paper and its Supplementary Information files or available on request from the corresponding author.

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

## Acknowledgements

This work was supported in part by Inserm, the Université Pierre et Marie Curie (UPMC, Paris 6), and an ERC advanced investigator grant (#250349) to J.-A.G. A.G. was partly supported by the King Abdullah University of Science and Technology (KAUST) Office of Sponsored Research award (#OSR-2015-CRG4-2602) to J.-A.G. and Stefan Arold. J.-C.P. lab is supported by grants from the Human Frontier Science Program (RGP0022/2013) and the Fondation pour la Recherche Médicale (DEQ20140329539). Q.C. and C.S. were recipients of doctoral fellowships of UPMC. Equipment at the IFM was also supported by DIM NeRF from Région Ile-de-France and by the FRC/Rotary 'Espoir en tête'. Microscopy was carried out at the Institut du Fer à Moulin Cell and Tissue Imaging facility. Labs of J.-A.G. and J.-C.P. are affiliated with the Paris School of Neuroscience (ENP) and the Bio-Psy Laboratory of excellence. Work in S.G. and J.A. labs was supported by Ministerio de Ciencia e Innovación (SAF2015-67474-R; MINECO/FEDER to S.G. and SAF2014-57160 to J.A.), Fundacio La Marato TV3, and Centro de Investigación Biomédica en Red sobre Enfermedades Neurodegenerativas (CIBERNED, R006/0010/0006). We thank Ana López and Maria Teresa Muñoz for technical assistance, and Teresa Rodrigo Calduch and the staff of the animal care facility (Facultat de Psicologia, Universitat de Barcelona) for their help.

## Author contributions

A.G. conceived and carried out most experiments, analysed and interpreted results and wrote the manuscript. V.B. carried out experiments related to HD and mouse model. Q.C. and Y.O. carried out and analysed electrophysiological experiments. C.S. carried out cell fractionation experiments. C.C.-D. carried out electron microscopy. B.d.P. and R.C. contributed to behavioural and biochemical experiments. J.A. and S.G. supervised Huntington-related experiments. J.-C.P. supervised and analysed electrophysiological experiments. J.-A.G. conceived and supervised the study, analysed results and wrote the manuscript.

## Additional information

**Competing interests:** The authors declare no competing financial interests.

