## [Peer Review File · Nature Communications]

Reviewers' Comments:

Reviewer #1 (Remarks to the Author)

This paper presents strong *in vivo* and *in vitro* evidence of an essential role for Pyk2 in hippocampal neuron spine formation and/or maintenance, and in synaptic function. The results also suggest that defects in Pyk2 function may be responsible for pathogenesis of hippocampal function in Huntington disease. The data are diverse and compelling, and include studies of behavior, receptor signaling, neuronal architecture in genetically modified animals, and LTP in hippocampal slices. The authors also report subcellular fractionation studies of postsynaptic densities, structural studies in cultured hippocampal neurons, and immunoelectron microscopy. Results are also validated using Cre-Lox technology to rule out a developmental effect associated with germ line defects in Pyk2. The results uniformly support a role for Pyk2 in hippocampal excitatory synapse function, and should greatly influence thinking in the field.

A surprising outcome is that the readouts obtained in behavioral and LTP studies were equivalent for heterozygous and homozygous Pyk2⁻ mice, while alterations in associated proteins and downstream substrates displayed proportional effects. The authors suggest a quantitative threshold for Pyk2 mediated synaptic and/or spine functions at the physiological or behavioral level.

Statistical analyses appear appropriate and valid. The only potential concern is with immunoelectron microscopy, which can distort morphology and possibly make discrimination of asymmetric and symmetric synapses difficult. Nonetheless, the results shown are quite significant, and consistent with other findings.

Another surprising finding was that Y402, a tyrosine residue associated with src recruitment and activation of Pyk2, was required for glutamate-induced PSD-95 synaptic translocation, but did not appear to have a role in mediating Pyk2's role in promoting spine density. The authors comment on this observation in the Discussion, proposing a scaffolding role for Pyk2. Are they suggesting that the kinase activity of the enzyme is not required for spine density effects, or is their interpretation that activation of Pyk2 in this context may proceed through an alternative pathway (e.g., not involving src recruitment). Finally, did they perform any experiments with the kinase dead Pyk2(K457A) mutant?

Other comments:

Lines 85-92: Studies of LTP were conducted in CA1 neurons, but the authors should mention whether they obtained results in other neurons or whether they simply restricted their study to CA1.

In Figure 6C, the spines from the knockout mice look shorter and less branched than those in the wild-type mice (as noted in the text and Fig. 3), even when expressing Pyk2 or Pyk2(Y402F). Were any measurements made of length (spine neck length) or branching in the Pyk2 or Pyk2(Y402) rescued mice?

While in the present study Y402 of Pyk2 did not have an essential role in spine density, there is other evidence that P-Y402-Pyk2 is redistributed away from the synapses in response to stimuli that reduce dendritic complexity. Consistent with the results in the present report, navigation of Pyk2 away from the synapses should negatively regulate spine density as well as other synaptic functions. The authors may wish to mention a study showing that redistribution of Pyk2(P-Y402) to the perinuclear region is observed in the hippocampus of chronically stressed rodents under conditions that have been shown to result in dendritic retraction (PNAS (2014) 111(45) 16130-5).

Some possible typographical/grammatical changes:

line 26: are under the

line 32: revealed autophosphorylation-dependent and

line 34: of the R6/1 mouse model

line 34: Normalizing Pyk2 levels in the hippocampus of 6/1 mice

line 41: spine morphology

line 47: a role for Pyk2

line 48: Ca²⁺ , and although the mechanism has not been fully elucidated, it probably involves dimer assembly, which triggers autophosphorylation at

line 62: of the Pyk2

line 70: in the HD

line 74: used a knockout

line 107: NMDA receptor subunits

line 110: indicating deficient (remove the "a")

line 111: In contrast, total GluN2A

line 135: clarify whether "this" protein refers to PSD-95 or Pyk2

line 140: using the Golgi-Cox method

line 160: mice as compared (remove the "showed")

line 210: of PSD-95. This interaction (remove the "while")

line 283: The precise contribution of GluN2A tyrosine phosphorylation to the synaptic alterations that increase NMDA receptor currents is difficult

line 294: the absence of LTP

Reviewer #2 (Remarks to the Author)

Giralt and colleagues report an extensive study that elucidates many functions of Pyk2 in synapse function, behaviour, physiology and disease. The fact that Pyk2 is calcium regulated has made it one of the most interesting synaptic kinases, yet its biochemical and physiological roles have not been extensively characterised. Here, the authors have made a major contribution with a wealth of interesting results. They have performed the first characterisation of Pyk2 knockout mice (and floxed mice) in the nervous system and confirm that it is essential for synaptic plasticity and learning and reveal a very interesting link to Huntington's disease. This reviewer commends them for the extensive and comprehensive analysis.

The authors present some potentially very important biochemical results, for which this reviewer would like to suggest a mechanistic interpretation. The changes in PSD-95 and GluN2A are dramatic as well as the phosphorylation changes in GluN2B. These findings may be linked by the fact that these three proteins coassemble into 1.5 KD "supercomplexes", which were recently described in Nature Communications (Frank et al DOI: 10.1038/ncomms11264). One of the key features of these supercomplexes was the tripartite assembly mechanism requires the GluN2B C-terminus with PSD-95 and PSD-93.

Giralt and colleagues show in Pyk2 mutants that the tyrosine phosphorylation on pY1472 of GluN2B C-terminus is diminished. Secondly, they show the site pY420 on Fyn is diminished indicating this kinase activity is reduced. Previous studies have shown that PSD-93 is a substrate for Fyn (Nada et al. J.Biol Chem, 2003). Thus, it would be interesting to test the following two hypotheses: i) is PSD-93 tyrosine phosphorylation diminished in Pyk2 mutant mice, and ii) are the 1.5 MDa supercomplexes altered. To test this latter point, the authors could run blue-native electrophoresis of extracts from Pyk2 mice and immunoblot with PSD-95 and NMDA receptor subunits (as described in Frank et al) as this would provide direct evidence for changes in the abundance of 1.5 MDa supercomplexes. They could also perform co-IP studies and immunoblots with antibodies to pY, PSD-95, PSD-93, GluN2B and GluN2A. It is also noteworthy that they report (Figure 5) glutamate-dependent Pyk2-dependent recruitment of PSD-95, which suggest the supercomplexes are activity-dependent, which has so far not been demonstrated in the literature.

This reviewer does not think that it is essential to perform these experiments for this manuscript.

The model that the NMDA-PSD95 supercomplexes (also referred to previously as MASCs or NMDA receptor complexes) are disrupted in Pyk2 mutants is also directly relevant to the behavioural and electrophysiological results. As summarised in Frank et al, many mutations have been described that disrupt these supercomplexes result in abnormal LTP and learning (including in GluN2B C-terminus, GluN2A, PSD95 and PSD-93). Hence, it has been proposed that these supercomplexes are key molecular machines underpinning these physiological and behavioural processes. Thus, I would recommend that they modify their discussion to indicate this link between Pyk2 and PSD-95/NMDA receptor supercomplexes.

Additional comments:

1. Synaptic physiology. The characterisation of the physiology is useful but minimal. For example, there is no characterisation of short-term plasticity (e.g. paired-pulse facilitation) which is typically found in studies of KO mice. Readers may be inclined to think that the plasticity deficits are specific to LTP and do not affect these short-term forms. Moreover, because the authors report that PSD95 and NMDA receptor biochemistry is altered, and mice carrying mutations in these genes have been reported to show changes in short term plasticity, it is likely that short-term plasticity changes will be observed. Many studies of mutant mice also include additional protocols for long-term plasticity such as different frequency-trains, spike-timing dependent plasticity and so on. These can be useful for teasing out quantitative effects of the stimulus trains, which may be important here, because the authors show heterozygous and homozygous mice have the same 100-Hz phenotype. This reviewer would prefer to see these experiments, however as an alternative, the authors should include a paragraph in the discussion about the limitations. I would expect that a detailed electrophysiology manuscript could be published in future.

2. Spine-density and PSD-95 puncta. In Figure 3, they report a reduction in PSD-95 puncta and show a high-magnification image of a small piece of CA1 region. They could show (in supplementary) a low-magnification picture of the hippocampus as there is considerable diversity of PSD-95 across regions and even within the CA1 (Broadhead et al, Sci. Reports 2016 and references therein).

3. Their study of Floxed mice and adult viral infection is impressive.

4. Huntington studies. Although they cite a paper showing this, I think it would be useful to have in Figure 7d/e, a panel probed with PSD-95. Especially since in Figure 7 f,g they show PSD-95 immunohistochemistry. Their rescue results are remarkable.

Seth Grant

Reviewer #3 (Remarks to the Author)

The authors present a relatively comprehensive study of the role of Pyk2 in hippocampal synaptic plasticity, spine maintenance and organization of postsynaptic density protein complexes. Although previous studies have implicated Pyk2 in the induction of LTP at CA3/CA1 synapses, this study significantly extends the work by using a Pyk2 knockout mouse to show the effects of Pyk2 deficiency on molecular organization of the PSD, induction of LTP, and hippocampal-dependent memory tasks. In addition, they show data to suggest that deficiency in Pyk2 expression in the hippocampus contributes to a deficit in hippocampal LTP and learning/memory in the R6/1 mouse

model of Huntington's disease.

In general, the paper is well-written and the data support most of the conclusions. I have a few concerns, however.

1) The Pyk2 KO is generated by deleting exons related to kinase function. Could this mouse line reflect a dominant negative effect, in which a truncated version of Pyk2 is expressed that can still interact with some proteins but has no kinase function? The authors should show whether any Pyk2 transcript up to exon 15 is expressed and whether an N-terminal fragment of Pyk2 protein is expressed.

2) The data from human brain tissue showing a significant reduction in Pyk2 protein levels is from Grade 3 and 4 brains; this is quite late-stage and does not address the issue of a potential role for Pyk2 deficiency in the early cognitive changes that occur in prodromal and early stage HD. In fact, most synaptic proteins show significantly reduced levels at this late stage of HD, because there is significant synapse loss.

3) The data on the R6/1 mouse, showing about 50% reduction of Pyk2 and then looking at the effects on NMDA receptor subunit expression/phosphorylation, seems to match with the previous data in Figure 2 showing what impact Pyk2^{+/-} has on those proteins. On the other hand, the R6/1 shows a robust reduction in PSD5 puncta density in hippocampus, whereas PSD95 protein levels are not significantly affected in mice with a 50% deficiency of Pyk2, and only the authors only show a reduction in PSD-95 protein levels and puncta density in Pyk2^{-/-} mice. These data suggest something other than a 50% reduction in Pyk2 is responsible for the reduction in PSD-95 puncta. Could the authors discuss this discrepancy?

4) Although the authors are able to show improvement in hippocampal-related memory tasks and PSD95 puncta density with AAV-mediated expression of Pyk2 in the hippocampus of R6/1 mice, I would still soften the conclusion that this is the critical mechanism underlying the deficit in LTP in hippocampal-dependent learning in these mice. First, there are a variety of other mechanisms that may be at play as these mice have a wide range of transcriptional changes, as well as receptor trafficking changes, previously documented by other groups. As well, the authors need to demonstrate that there is an LTP deficit in the R6/1 mice in their hands at this age, and second, that it's rescued by expression of the AAV-Pyk2 construct in the hippocampus, to match with their behavioral and molecular results.

5) In Figure 2 the large discrepancy between total GluN2B (no change) and GluN2B in the PSD (reduced by 75%) in the Pyk2^{-/-} vs. Pyk2^{+/+} mice hippocampal tissue seems surprising. It is established that GluN1 is expressed in excess and that a significant proportion does not contribute to functional NMDAR complexes; therefore, one would expect to see a large proportion outside of the PSD and that might explain the observation of no difference in total GluN1 whereas there is a significant difference in the PSD expression. However, both GluN2A and GluN2B largely reside in the PSD in mature hippocampal neurons, so it seems unlikely that a 75% decrease in GluN2B in the PSD would not be reflected in total expression, unless the total excludes the PSD fraction because of detergent conditions? But then why does the GluN2A expression show a good match between PSD and total expression? If the mice are young (1 – 2 weeks old) and a good proportion of the GluN2B is not in the PSD that might explain the results. How old are the mice used in these experiments? The age of mice used in each set of experiments should be clearly stated in the figure legends and/or methods.

6) Figure 3: It seems unusual that the decrease of PSD95 puncta is approximately double the decrease in spine density; it might suggest more immature spines in the Pyk2^{-/-} mice, but such immature spines usually have longer necks and smaller heads (i.e. filopodia), which is not supported by the data. Could the authors discuss this point?

7) Figure 4D: correct the labeling of the Y-axis.

8) Figure 7: Since novel location or open arm exploration may reflect more than just hippocampal contributions, do the authors find Pyk2 expression extending into the cortex after the AAV injection? This should be made clear in the figures.

General changes to the manuscript:

In the preparation of the revised version we changed/improved a number of things to comply with Nat Commun instructions and/or to correct errors we identified (in addition to changes to answer Referees reviews, see below).

- We reviewed all the statistical analysis and carried out a normality check. When the variable distribution was significantly different from normal, we used a non-parametric test (Mann and Whitney or Kruskal and Wallis + Dunn's post hoc test for multiple comparisons). This normality check had not been done for all data sets previously, but was generalized in the revised version. Changes were done when appropriate, i.e. for some proteins in Fig. 2 and Supplementary Fig. 2, and for Supplementary Figures 1a, 1b, 3b, 3c, and 3d. The detailed additional statistical analysis for Fig. 2 and Supplementary Fig. 2 is included in modified Supplementary Table 1. The change of test modified the p values obtained with the non-parametric tests as compared to previously used ones, but did not lead to changes in conclusions.

- We corrected the errors we identified in some figures. In Fig. 2f bars for GluN2A and GluN2B had been inadvertently inverted and were corrected. In Fig. 3g the correct °° was put instead of °. In Fig. 5 tubulin blot, in Supplementary Fig. 2d pGluA1 and pGluA2 blots, the pictures were replaced by less fuzzy ones. In Supplementary Fig. 4, NeuN blot was erroneous and replaced by the correct one.

- We included additional Supplementary Figs. 5 to 14, to display the full gels/immunoblots shown in figures. We also added vertical lines in blot excerpts shown in figures, to clearly identify the immunoblots extracts, which were not run side by side, when this was the case.

- Axes labeling in figures has been modified when necessary to fit Nat Commun requirements.

- We corrected a few typos and improved some sentences in addition to those signaled by Referees.

- We changed the wording of some sentences in the Abstract, of Results subtitles, and some figure legends to remain within the requested length (150 words for abstract, 60 characters including space for subtitles, and 350 words for figure legends).

All significant modifications in the text are highlighted yellow.

We have carried out additional experiments as requested by Referees (see details in the point-by-point responses). These are shown as:

- Fig. 1e
- Fig. 2e
- Fig. 3e
- Fig. 6a and c, images added for new experiments for Pyk2(KD)
- Fig. 6b, d, and e: new experiments including Pyk2(KD) and the other forms have been carried out and the results pooled with the previous experiments.
- Fig. 7d and e: blots for PSD-95 have been carried out, quantified, and added to the figure.
- Fig. 8e and f.
- Suppl. Fig. 1e
- Suppl. Fig. 2a
- Suppl. Fig. 3e
- Suppl. Fig. 4a and b, 4f (lower magnification pictures).

Point by point responses to the Referees' comments:

Reviewer #1 (Remarks to the Author):

This paper presents strong in vivo and in vitro evidence of an essential role for Pyk2 in hippocampal neuron spine formation and/or maintenance, and in synaptic function. The results also suggest that defects in Pyk2 function may be responsible for pathogenesis of hippocampal function in Huntington disease. The data are diverse and compelling, and include studies of behavior, receptor signaling, neuronal architecture in genetically modified animals, and LTP in hippocampal slices. The authors also report subcellular fractionation studies of postsynaptic densities, structural studies in cultured hippocampal neurons, and immunoelectron microscopy. Results are also validated using Cre-Lox technology to rule out a developmental effect associated with germ line defects in Pyk2. The results uniformly support a role for Pyk2 in hippocampal excitatory synapse function, and should greatly influence thinking in the field.

A surprising outcome is that the readouts obtained in behavioral and LTP studies were equivalent for heterozygous and homozygous Pyk2- mice, while alterations in associated proteins and downstream substrates displayed proportional effects. The authors suggest a quantitative threshold for Pyk2 mediated synaptic and/or spine functions at the physiological or behavioral level.

Statistical analyses appear appropriate and valid. The only potential concern is with immunoelectron microscopy, which can distort morphology and possibly make discrimination of asymmetric and symmetric synapses difficult. Nonetheless, the results shown are quite significant, and consistent with other findings.

We thank the Referee for his/her positive comments. We agree with the potential difficulties with electron microscopy. Nevertheless, we would like to stress that the proportion of symmetric and asymmetric synapses that we measured in the *stratum radiatum* are in the same range as those reported in the literature (e.g., Megias et al. *Neuroscience*, 2001, 102:527-40).

Another surprising finding was that Y402, a tyrosine residue associated with src recruitment and activation of Pyk2, was required for glutamate-induced PSD-95 synaptic translocation, but did not appear to have a role in mediating Pyk2's role in promoting spine density. The authors comment on this observation in the Discussion, proposing a scaffolding role for Pyk2. Are they suggesting that the kinase activity of the enzyme is not required for spine density effects, or is their interpretation that activation of Pyk2 in this context may proceed through an alternative pathway (e.g., not involving src recruitment). Finally, did they perform any experiments with the kinase dead Pyk2(K457A) mutant?

To address this highly relevant question, we repeated the *in vitro* cell culture experiments including a kinase-dead Pyk2 construct (Pyk2-KD) in addition to the other constructs. With this KD mutant the results were similar to those obtained with the Y402F mutation, with a rescue of spine density but not of PSD-95 puncta increase after glutamate treatment (results now included in Figure 6a-d, which pool all this series of experiments, and in Results section, p.8). Taken together these results concur to suggest that the role of Pyk2 in spine density in this context could be mediated through an alternative pathway (not involving Src recruitment nor Pyk2 kinase activity), including a putative scaffolding role of Pyk2 that is able to interact with many protein partners. Interestingly such a kinase/autophosphorylation independent role has been previously identified for the related Tyr kinase FAK (Corsi et al., *J Biol Chem.* 2009, 84:34769-76; Zhao X, et al. *J Cell Biol.* 2010 189:955-65). The corresponding sentences in the Discussion have been modified to better discuss this possibility, p. 13):

"Dendritic spine density and length were also altered in Pyk2 mutant mice. Our study of the requirement of Pyk2 protein domains and functions for its various roles reveals the complexity of

its contribution. The kinase activity of Pyk2 and its autophosphorylation site, Tyr402, which is critical for the recruitment of SFKs, were both necessary for the rescue of PSD-95 clustering in spines in Pyk2 KO neurons. The C-terminal region, which allows interaction with PSD-95¹⁴, was also required for PSD-95 clustering. In contrast, rescue of spine density required the C-terminal region but neither Tyr402 nor the kinase activity, indicating a phosphorylation-independent role of Pyk2 in the regulation of spine number. Such an autophosphorylation/kinase activity-independent function of Pyk2 is reminiscent of those reported for the closely related FAK^{52,53} in non-neuronal cells. This alternate signaling is presumably SFK-independent and may be linked to scaffolding properties of Pyk2 and/or its interaction with specific partners.”

Other comments:

Lines 85-92: Studies of LTP were conducted in CA1 neurons, but the authors should mention whether they obtained results in other neurons or whether they simply restricted their study to CA1.

We agree with the referee that it will be very interesting to study other hippocampal neurons or other brain regions in order to have a wider view of Pyk2. However in this first study we focused on CA1 pyramidal neurons. This is now indicated p. 4 “We restricted our study to CA1,…”.

In Figure 6C, the spines from the knockout mice look shorter and less branched than those in the wild-type mice (as noted in the text and Fig. 3), even when expressing Pyk2 or Pyk2(Y402F). Were any measurements made of length (spine neck length) or branching in the Pyk2 or Pyk2(Y402) rescued mice?

In order to address the referee's query we evaluated spine length in all the *in vitro* conditions (including re-expression of KD Pyk2, see above) and we observed that in all of them the spines were shorter as compared to wild type controls, as already observed in tissue sections (Fig. 3i). We have included these new results in Fig. 6e and in the Results section p. 8) “Transfection of GFP:Pyk2₁₋₈₄₀ had no significant effect, but, in contrast to what we observed for PSD-95 puncta rescue (see Fig. 6b), both GFP:Pyk2_{Y402F} and GFP:Pyk2-KD fully restored spine density (Fig. 6c, d), revealing a role for Pyk2 independent of its autophosphorylation and kinase activity. We also quantified the effects of Pyk2 deletion on spine length (Fig. 6e). In the absence of Pyk2, spines were shorter, as observed *in vivo* (see Fig. 3i), but this effect was not rescued by re-expression of wild type or mutated Pyk2 (Fig. 6e). This lack of rescue may indicate a contribution of presynaptic Pyk2 in spine length regulation since with the low transfection rate in our culture system, concomitant transfection of pre- and post-synaptic neurons was very rare.”

We discuss these results in the Discussion section p. 13: “Finally, in the absence of Pyk2 spine length was decreased both *in vivo* and *in culture*. A possible explanation for this lack of rescue could be a dual role of Pyk2 at the pre- and post-synaptic levels, which were not simultaneously restored at the same synapses in the culture conditions in which the transfection rate was low. At any rate, the mechanisms by which Pyk2 controls spine length are likely to be complex since both positive and negative effects of Pyk2 and FAK on spine growth have been previously reported⁵⁸⁻⁶⁰ and their elucidation will require further investigation. The important aspect of the present results is the overall deficit in spine number and length in the absence of Pyk2.”

While in the present study Y402 of Pyk2 did not have an essential role in spine density, there is other evidence that P-Y402-Pyk2 is redistributed away from the synapses in response to stimuli that reduce dendritic complexity. Consistent with the results in the present report, navigation of Pyk2 away from the synapses should negatively regulate spine density as well as other synaptic functions. The authors may wish to mention a study showing that redistribution of Pyk2(P-Y402) to the perinuclear region is observed in the hippocampus of chronically stressed rodents under conditions that have been shown to result in dendritic retraction (PNAS (2014) 111(45) 16130-5).

We thank the Referee for this interesting suggestion about the possible role of Pyk2 redistribution on synaptic properties and we now mention this possibility in the Discussion section p. 13:

“Interestingly, it has been shown that chronic stress induces a redistribution of activated Pyk2 to the perinuclear region of CA3 neurons, contributing to a deficit in the nuclear pore protein NUP62 and its potential negative consequences on dendritic complexity⁶¹. The present study suggests that Pyk2 redistribution could also directly contribute to dendritic or synaptic alterations by reducing its local levels.”

Some possible typographical/grammatical changes:

line 26: are under the

line 32: revealed autophosphorylation-dependent and

line 34: of the R6/1 mouse model

line 34: Normalizing Pyk2 levels in the hippocampus of 6/1 mice

line 41: spine morphology

line 47: a role for Pyk2

line 48: Ca²⁺ , and although the mechanism has not been fully elucidated, it probably involves dimer assembly, which triggers autophosphorylation at

line 62: of the Pyk2

line 70: in the HD

line 74: used a knockout

line 107: NMDA receptor subunits

line 110: indicating deficient (remove the "a")

line 111: In contrast, total GluN2A

line 135: clarify whether "this" protein refers to PSD-95 or Pyk2

line 140: using the Golgi-Cox method

line 160: mice as compared (remove the "showed")

line 210: of PSD-95. This interaction (remove the "while")

line 283: The precise contribution of GluN2A tyrosine phosphorylation to the synaptic alterations that increase NMDA receptor currents is difficult

line 294: the absence of LTP

We thank the Referee for pointing out these errors, which have been corrected in the manuscript. Line 283 we have modified the whole sentence to try and better express what we originally meant.

Reviewer #2 (Remarks to the Author):

Giralt and colleagues report an extensive study that elucidates many functions of Pyk2 in synapse function, behaviour, physiology and disease. The fact that Pyk2 is calcium regulated has made it one of the most interesting synaptic kinases, yet its biochemical and physiological roles have not been extensively characterised. Here, the authors have made a major contribution with a wealth of interesting results. They have performed the first characterisation of Pyk2 knockout mice (and floxed mice) in the nervous system and confirm that it is essential for synaptic plasticity and learning and reveal a very interesting link to Huntington's disease. This reviewer commends them for the extensive and comprehensive analysis.

The authors present some potentially very important biochemical results, for which this reviewer would like to suggest a mechanistic interpretation. The changes in PSD-95 and GluN2A are dramatic as well as the phosphorylation changes in GluN2B. These findings may be linked by the fact that these three proteins coassemble into 1.5 KD "supercomplexes", which were recently described in Nature Communications (Frank et al DOI: 10.1038/ncomms11264). One of the key features of these supercomplexes was the tripartite assembly mechanism requires the GluN2B C-terminus with PSD-95 and PSD-93.

Giralt and colleagues show in Pyk2 mutants that the tyrosine phosphorylation on pY1472 of GluN2B C-terminus is diminished. Secondly, they show the site pY420 on Fyn is diminished indicating this kinase activity is reduced. Previous studies have shown that PSD-93 is a substrate for Fyn (Nada et al. J.Biol Chem, 2003). Thus, it would be interesting to test the following two hypotheses: i) is PSD-93 tyrosine phosphorylation diminished in Pyk2 mutant mice, and ii) are the 1.5 MDa supercomplexes altered. To test this latter point, the authors could run blue-native electrophoresis of extracts from Pyk2 mice and immunoblot with PSD-95 and NMDA receptor subunits (as described in Frank et al) as this would provide direct evidence for changes in the abundance of 1.5 MDa supercomplexes. They could also perform co-IP studies and immunoblots with antibodies to pY, PSD-95, PSD-93, GluN2B and GluN2A. It is also noteworthy that they report (Figure 5) glutamate-dependent Pyk2-dependent recruitment of PSD-95, which suggest the supercomplexes are activity-dependent, which has so far not been demonstrated in the literature. This reviewer does not think that it is essential to perform these experiments for this manuscript.

We thank the Referee for his positive comments and suggestions. We agree that the suggested experiments are very interesting and will be the object of future work.

The model that the NMDA-PSD95 supercomplexes (also referred to previously as MASCs or NMDA receptor complexes) are disrupted in Pyk2 mutants is also directly relevant to the behavioural and electrophysiological results. As summarised in Frank et al, many mutations have been described that disrupt these supercomplexes result in abnormal LTP and learning (including in GluN2B C-terminus, GluN2A, PSD95 and PSD-93). Hence, it has been proposed that these supercomplexes are key molecular machines underpinning these physiological and behavioural processes. Thus, I would recommend that they modify their discussion to indicate this link between Pyk2 and PSD-95/NMDA receptor supercomplexes.

We thank the Referee for this suggestion that was very helpful to improve the Discussion section. We have now modified it to indicate the possibility that a putative consequence of the absence of Pyk2 is a severe alteration of NMDAR-PSD95 supercomplexes on the basis of the work of Frank et al. Discussion section, p. 12:

"Our study shows that Pyk2 is required for the later recruitment of PSD-95 to spines revealing its contribution in the coordinated Ca²⁺-dependent dynamics of PSD proteins, a key aspect of synaptic function and plasticity. Importantly, GluN2A, GluN2B, and PSD-95 co-assemble together with PSD-93 into 1.5 MDa "supercomplexes"⁴⁸. PSD-93 is phosphorylated by Fyn⁴⁹, which is altered in PyK2 mutant mice. Since both PSD-93 and PSD-95 promote Fyn-mediated tyrosine phosphorylation of GluN2A and/or GluN2B^{50,51}, it appears that Pyk2 is potentially located at a strategic location to regulate NMDA receptor supercomplexes. Since these supercomplexes have been proposed to be functionally important, with mutations in their key components resulting in abnormal LTP and learning⁴⁸, it will be particularly interesting to investigate whether their alteration accounts for LTP

and other synaptic deficits in Pyk2 mutant mice.”

Additional comments:

1. Synaptic physiology. The characterisation of the physiology is useful but minimal. For example, there is no characterisation of short-term plasticity (e.g. paired-pulse facilitation) which is typically found in studies of KO mice. Readers may be inclined to think that the plasticity deficits are specific to LTP and do not affect these short-term forms. Moreover, because the authors report that PSD95 and NMDA receptor biochemistry is altered, and mice carrying mutations in these genes have been reported to show changes in short term plasticity, it is likely that short-term plasticity changes will be observed. Many studies of mutant mice also include additional protocols for long-term plasticity such as different frequency-trains, spike-timing dependent plasticity and so on. These can be useful for teasing out quantitative effects of the stimulus trains, which may be important here, because the authors show heterozygous and homozygous mice have the same 100-Hz phenotype. This reviewer would prefer to see these experiments, however as an alternative, the authors should include a paragraph in the discussion about the limitations. I would expect that a detailed electrophysiology manuscript could be published in future.

We fully agree with the Referee that it will be extremely interesting to carry out an extensive characterization of synaptic function and plasticity in Pyk2 mutant mice. The current study was, however, not oriented towards an extensive characterization of electrophysiological alterations. We added a sentence, as requested, p. 12, to indicate this limitation: “Many other aspects of synaptic function and plasticity remain to be investigated in Pyk2 mutant mice ...”.

To start addressing short-term plasticity, we compared paired-pulse ratio in the 3 genotypes to evaluate paired-pulse facilitation. We observed that paired-pulse facilitation was very markedly decreased in Pyk2 mutant mice. Since this form of short term plasticity is thought to result from presynaptic mechanisms, it indicates a role of Pyk2 in its regulation in presynaptic terminals. This is in agreement with the existence of some presynaptic Pyk2 (current **Fig. 3a**, **Suppl. Fig. 3a**, and Bongiorno-Borbone et al. 2002, J Neurochem 81:1212-22). It is also possible that post-synaptic mechanisms contribute to this alteration as suggested for some forms of PPF (e.g. Yang, S., et al. Front Cell Neurosci, 2016, 10:224). We have added the PPR results as new **Fig. 1e** and **Suppl. Fig. 1e**, and in the Results section, p.4-5 “We also examined a form of short-term plasticity at the same synapses. Paired-pulse facilitation was observed in wild-type mice but was markedly decreased in both homozygous and heterozygous Pyk2 mutant mice (**Fig. 1e**, **Supplementary Fig. 1e**), suggesting a presynaptic role for Pyk2.” We also discuss these new results p. 13: “The current study also suggests a role for Pyk2 at the presynaptic level. Our electron microscopy experiments show the presence of Pyk2 in nerve terminals, confirming previous biochemical observations⁵⁴. Its functional role is indicated by the alteration in paired-pulse facilitation, a form of short-term plasticity that is considered to be mostly presynaptic^{55,56}, although post-synaptic mechanisms are also possible⁵⁷.”. We thank the Referee for suggesting us to address this point in this manuscript.

2. Spine-density and PSD-95 puncta. In Figure 3, they report a reduction in PSD-95 puncta and show a high-magnification image of a small piece of CA1 region. They could show (in supplementary) a low-magnification picture of the hippocampus as there is considerable diversity of PSD-95 across regions and even within the CA1 (Broadhead et al, Sci. Reports 2016 and references therein).

We have added in the supplementary material representative low-magnification mosaic pictures spanning the full depth of CA1 immunostained for PSD-95, in Pyk2+/+, Pyk2+/- and Pyk2-/- mice (**Supplementary Fig. 3e**).

3. *Their study of Floxed mice and adult viral infection is impressive.*

We thank the Referee for his positive comment.

4. *Huntington studies. Although they cite a paper showing this, I think it would be useful to have in Figure 7d/e, a panel probed with PSD-95. Especially since in Figure 7 f,g they show PSD-95 immunohistochemistry. Their rescue results are remarkable.*

We have now performed an immunoblot experiment to measure PSD-95 levels and we confirm the decrease in PSD-95 protein levels previously reported in R6/1 mice, new results added in **Fig. 7d** and **7e**, and in the corresponding Results section text p. 9, bottom.

Reviewer #3 (Remarks to the Author):

The authors present a relatively comprehensive study of the role of Pyk2 in hippocampal synaptic plasticity, spine maintenance and organization of postsynaptic density protein complexes. Although previous studies have implicated Pyk2 in the induction of LTP at CA3/CA1 synapses, this study significantly extends the work by using a Pyk2 knockout mouse to show the effects of Pyk2 deficiency on molecular organization of the PSD, induction of LTP, and hippocampal-dependent memory tasks. In addition, they show data to suggest that deficiency in Pyk2 expression in the hippocampus contributes to a deficit in hippocampal LTP and learning/memory in the R6/1 mouse model of Huntington's disease.

We thank the Referee for his/her positive comments.

In general, the paper is well-written and the data support most of the conclusions. I have a few concerns, however.

1) *The Pyk2 KO is generated by deleting exons related to kinase function. Could this mouse line reflect a dominant negative effect, in which a truncated version of Pyk2 is expressed that can still interact with some proteins but has no kinase function? The authors should show whether any Pyk2 transcript up to exon 15 is expressed and whether an N-terminal fragment of Pyk2 protein is expressed.*

We agree with the Referee that this is an important point. Thus, to address it we have used an antibody that interacts with the first 1-100 amino acids of Pyk2 (Abcam rabbit anti-Pyk2 #ab32571). There was no evidence for a Pyk2 N-terminal domain expression in the Pyk2 knockout mice. This result demonstrates that the partially deleted mRNA and/or the truncated Pyk2 protein are not stable in the knockout mutant and that no N-terminal fragment is involved in the observed phenotype. This is shown in new **Supplementary Fig. 2a** and mentioned in the Results p. 5: "No N-terminal truncated fragment was detected in the knockout mice (**Supplementary Fig. 2a**), showing that deletion of exons 15-18 in the Pyk2 gene²⁰ destabilized the resulting mRNA and/or protein."

2) *The data from human brain tissue showing a significant reduction in Pyk2 protein levels is from Grade 3 and 4 brains; this is quite late-stage and does not address the issue of a potential role for Pyk2 deficiency in the early cognitive changes that occur in prodromal and early stage HD. In fact, most synaptic proteins show significantly reduced levels at this late stage of HD, because there is significant synapse loss.*

We agree with the Referee and have carried out new experiments. In order to test whether Pyk2 has a potential role in early cognitive changes that occur in prodromal and early stages of the disease, we measured Pyk2 levels in 4 new postmortem samples of patients with grade 1-2 HD. Pyk2 protein levels were not significantly altered in these samples (**Supplementary Fig. 4a, b**) indicating that Pyk2 levels alterations are more

likely to play a role in intermediate-late stages of HD than at early stages. The corresponding results have been added in the Results section, p. 9: "In patients with intermediate or late HD (stage 3-4) Pyk2 levels were reduced to $64 \pm 8\%$ of controls (mean \pm SEM, **Fig. 7a, b**). In contrast in patients with prodromal or early stage (1-2) there was no significant change (**Supplementary Fig. 4a, b**)." The Discussion has also been adapted to include these new conclusions "softening" the potential role of Pyk2 in HD (see below).

3) The data on the R6/1 mouse, showing about 50% reduction of Pyk2 and then looking at the effects on NMDA receptor subunit expression/phosphorylation, seems to match with the previous data in Figure 2 showing what impact Pyk2^{+/-} has on those proteins. On the other hand, the R6/1 shows a robust reduction in PSD5 puncta density in hippocampus, whereas PSD95 protein levels are not significantly affected in mice with a 50% deficiency of Pyk2, and only the authors only show a reduction in PSD-95 protein levels and puncta density in Pyk2^{-/-} mice. These data suggest something other than a 50% reduction in Pyk2 is responsible for the reduction in PSD-95 puncta. Could the authors discuss this discrepancy?

We thank the Referee for this important question. Indeed, by immunoblotting we observed a tendency of PSD-95 to decrease in Pyk2^{+/-} mice as compared to Pyk2^{+/+} mice which do not reach significance. Therefore, to better evaluate synaptic PSD-95 we quantified PSD-95 puncta in Pyk2^{+/-} mice (new **Fig. 3e**). This quantification in the *stratum radiatum* of CA1 revealed a significant decrease in the number of PSD-95-positive particles in Pyk2^{+/-} mice as compared to Pyk2^{+/+} mice. Thus, we conclude that although they may be masked in total tissue extracts because of overall variability, the changes in synaptic PSD-95, which is the fraction presumably relevant for its function, are quite similar to those that occur in R6/1 mice in CA1. Results have been included (**Fig. 3e** and **Supplementary Fig. 3e**) and included in the Results section p.6: "The number of PSD-95-positive puncta was significantly reduced in Pyk2^{+/-} and even more so in Pyk2^{-/-} as compared to Pyk2^{+/+} mice (**Fig. 3d, e**), in agreement with the corresponding decrease in PSD-95 protein levels. This effect appeared consistent throughout CA1 depth (**Supplementary Fig. 3e**)." At any rate a similar reduction in PSD-95 puncta in R6/1 and Pyk2^{+/-} mice does not prove that the decrease in Pyk2 is responsible for the alteration in PSD-95 puncta. This is why we carried out the AAV Pyk2 expression experiments. The correction of PSD-95 puncta in R6/1 mice following AAV Pyk2 (**Fig. 8i-k**) provides strong evidence for the causal role of Pyk2 deficit.

4) Although the authors are able to show improvement in hippocampal-related memory tasks and PSD95 puncta density with AAV-mediated expression of Pyk2 in the hippocampus of R6/1 mice, I would still soften the conclusion that this is the critical mechanism underlying the deficit in LTP in hippocampal-dependent learning in these mice. First, there are a variety of other mechanisms that may be at play as these mice have a wide range of transcriptional changes, as well as receptor trafficking changes, previously documented by other groups. As well, the authors need to demonstrate that there is an LTP deficit in the R6/1 mice in their hands at this age, and second, that it's rescued by expression of the AAV-Pyk2 construct in the hippocampus, to match with their behavioral and molecular results.

We agree with the Referee that although we show improvements in hippocampal-related memory tasks and PSD95 puncta density with AAV-mediated expression of Pyk2 in the hippocampus of R6/1 mice, many other alterations in the disease may contribute to the LTP deficit. In fact we are sorry if our text gave that erroneous impression since it is not what we meant. Thus, we have carefully rephrased the corresponding sentences in the main text (Introduction, Discussion) when referring to the possible role of Pyk2 in HD symptoms to avoid misinterpretation/over interpretation. We think that the interesting aspect of our study with respect to HD is that Pyk2 deficiency appears to contribute to some of the phenotype and that this contribution is partly reversible in the rodent model

upon Pyk2 re-expression. Of course there are many other mechanisms involved in the human disease or in the mouse model.

The Referee asks about the LTP deficit in R6/1 mice in our hands at this age and its possible rescue by AAV-Pyk2. We addressed this question experimentally. We injected AAV-GFP and AAV-Pyk2 in R6/1 mice as well as AAV-GFP in wild type mice as a control group, all mice being at the same age as those in which the behavioral experiments were carried out (4-5 months). We first tested whether LTP was inducible in AAV-GFP-injected wt mice at this age and found it was well detected in our experimental conditions. In R6/1 mice synaptic potentiation was observed immediately after high frequency stimulation, but rapidly decayed and fEPSPs were back to control levels over the last ten minutes of recording (new **Fig. 8e**). Thus, there was no LTP in the two R6/1 groups (injected with either AAV-GFP or AAV-Pyk2) compared to wt mice injected with AAV-GFP (new **Fig. 8f**). These results are described p. 10: "We also examined LTP in these mice at the same age as for behavioral experiments (4-5 months). In WT-GFP mice we observed a robust LTP in CA1 after stimulation of Schaffer collaterals (**Fig. 8e, f**). In contrast, in R6/1-GFP and R6/1-Pyk2 mice synaptic potentiation was not stable (**Fig. 8e, f**). One hour after HFS, potentiation was observed only in WT-GFP mice (**Fig. 8f**), revealing that restoration of Pyk2 levels was not sufficient to correct the LTP impairment." It should be noted that a lack of perfect matching between LTP in slices and behavioral studies has been previously reported in many publications in various models (including spatial memory and LTP in hippocampus, e.g., Meiri et al., 1998 PNAS, 95:15037-42 ; Reisel et al., 2002 Nat Neurosci. 59:868-73; Okabe et al., 1998 J Neurosci, 18:4177–4188).

5) In Figure 2 the large discrepancy between total GluN2B (no change) and GluN2B in the PSD (reduced by 75%) in the Pyk2^{-/-} vs. Pyk2^{+/+} mice hippocampal tissue seems surprising. It is established that GluN1 is expressed in excess and that a significant proportion does not contribute to functional NMDAR complexes; therefore, one would expect to see a large proportion outside of the PSD and that might explain the observation of no difference in total GluN1 whereas there is a significant difference in the PSD expression. However, both GluN2A and GluN2B largely reside in the PSD in mature hippocampal neurons, so it seems unlikely that a 75% decrease in GluN2B in the PSD would not be reflected in total expression, unless the total excludes the PSD fraction because of detergent conditions? But then why does the GluN2A expression show a good match between PSD and total expression? If the mice are young (1 – 2 weeks old) and a good proportion of the GluN2B is not in the PSD that might explain the results. How old are the mice used in these experiments? The age of mice used in each set of experiments should be clearly stated in the figure legends and/or methods.

We apologize because we did not specify the age of the mice in each experiment/result. We have now included the age in each figure legend. In summary, experiments in Pyk2 knockout or floxed mice were carried out at the age of ~12 weeks except for the LTP experiment in which mice were 3-4 weeks old. The experiments conducted in R6/1 mice were at ~20 weeks of age.

Regarding the discrepancy between total and synaptic GluN2B levels, tissue fractionation was done to detect a localized change in NMDA receptors subunits distribution, in spite of an absence of differences in total lysates, which could reflect an alteration in their targeting or turnover. The extracts for measuring total proteins were done by sonication of tissue in 1% SDS at ~100°C. We agree with the remarks of the Referee concerning the distribution of NMDA receptor subunits in wild type mice. However our results suggest that there is an alteration of this distribution in mice lacking Pyk2, which may modify the proportions expected from wild type mice, possibly as a consequence of impaired Tyr phosphorylation of NMDA receptor subunits and/or as an indirect consequence of alterations in other proteins such as PSD-95. In the case of

GluN2A, decreased total levels suggest that an additional mechanism is involved such as a higher degradation rate or, possibly, a decreased synthesis. It will be interesting in the future to investigate further these various possibilities and study in details the turnover rates and intracellular targeting of NMDA receptor subunits in the absence of Pyk2, but we thought it was outside the scope of the present manuscript. We have added further discussion of this issue in the Discussion section about the putative role of Pyk2 in the regulation of NMDARs-PSD-95 macrocomplexes as suggested by Referee #2 (Discussion, p. 12).

6) Figure 3: It seems unusual that the decrease of PSD95 puncta is approximately double the decrease in spine density; it might suggest more immature spines in the Pyk2^{-/-} mice, but such immature spines usually have longer necks and smaller heads (i.e. filopodia), which is not supported by the data. Could the authors discuss this point?

We thank the Referee for pointing out this discrepancy that requires some clarification. Not all spines are positive for PSD-95 (Isshiki et al., Nat Com 2014, 5:4742; Cane et al., J Neurosci 2014, 34:2075–86). Moreover, immunohistofluorescence detection of PSD-95 puncta has a threshold below which “puncta” are not identifiable, whereas morphological identification of spines has no real threshold. There may be a number of spines with low amounts of PSD-95 and perhaps, at least in some of them, replacement by related molecules.

Differences between quantification of spines and PSD-95 puncta in HD models have been previously reported in the same R6/1 mouse model with a decrease of ~15% in spine density by using Golgi staining and a decrease of ~50% of total PSD-95 and PSD-95-positive puncta in the same mice (Miguez et al., Hum Mol Genet, 2015 24:4958–70HMG; Anglada-Huguet et al., Neurobiol Dis 2016, 95:22–34). A large discrepancy between changes in PSD-95 or puncta is also apparent in reports on medium spiny neurons in the same HD transgenic model (Torres-Peraza et al., Neurobiol Dis, 2008, 29:409–21, Spires et al., Eur J Neurosci 2004, 19:2799-807). Discrepancy in proportionality between changes in spines and in PSD-95 clusters have also been reported in other brain regions and models (Rivera et al., Synapse 2013, 67:897–908). We have added a sentence to take this point into account p. 6: “The decrease in spine number was less pronounced than the decrease in PSD-95 puncta, possibly due to an immunofluorescence detection threshold and/or an increased number of spines lacking PSD-95.”

7) Figure 4D: correct the labeling of the Y-axis.

We thank the Referee for noticing this error. The Y-axis labeling in Fig. 4d has been corrected.

8) Figure 7: Since novel location or open arm exploration may reflect more than just hippocampal contributions, do the authors find Pyk2 expression extending into the cortex after the AAV injection? This should be made clear in the figures.

To address the Referee's concern about a widespread infection/transduction of other neuronal populations beyond hippocampus, we have added in the supplementary material (**Supplementary Fig. 4f**) representative images from the three groups showing the entire hippocampus and surrounding cortical areas. We did not observe significant infection/transduction in non-hippocampal regions using AAV-GFP or AAV-Pyk2. Therefore the observed rescue is mediated by increased Pyk2 expression in the hippocampus of R6/1 injected with AAV-Pyk2, which thus appears sufficient to improve the behavioral phenotype, even though other brain regions are also implicated in the corresponding normal behavior.

Reviewers' Comments:

Reviewer #1 (Remarks to the Author)

The authors have done an excellent job of responding to the questions raised in the reviews, and have modified and edited the manuscript accordingly. They have even performed additional experiments that address the reviewers' concerns and surpass expectations. This is a timely and important report and is ready for publication. It was a pleasure to review such exceptional work.

Reviewer #2 (Remarks to the Author)

The authors have addressed my concerns and the paper is suitable for publication.

Reviewer #3 (Remarks to the Author)

The paper is significantly stronger after revisions, which have addressed almost all of my concerns. This is an impressive study of the role of Pyk2 in hippocampal CA1 synaptic structure and plasticity. My only remaining concern is the over-statement of a role for Pyk2 deficit in cognitive impairment in HD, which is detailed below along with a few minor corrections/suggestions.

1. The title should be modified. It is too strong to say that Pyk2 reduction in hippocampus (which is all that is studied in this paper) "mediates Huntington's disease cognitive deficits". The cognitive deficits in HD are numerous and diverse, and mainly involve frontal executive dysfunction, especially in early stages of disease. Results in this manuscript are restricted to demonstrating rescue of two hippocampal-dependent memory tasks in one mouse model of HD. Therefore, please soften the statement in the title; instead of "mediates" could say "contributes to".

2. Along the same lines as #1, on pg. 13, line 392 it is misleading to talk about the decrease in Pyk2 in Grade 3-4 HD brains as supporting a role for Pyk2 decrease in HD pathophysiology. The strongest data in this manuscript is from the R6/1 mouse model of HD, whereas the results from HD patient brains must be discussed in a more balanced way, mentioning that in Grade 3-4 HD brains there is significant synaptic loss so that many synaptic proteins show reductions; moreover, the fact that at earlier stages (Grade 1-2) there was no Pyk2 reduction suggests that the role of Pyk2 decrease in hippocampal-dependent cognitive deficits in HD patients remains to be determined.

3. Again, on pg. 11, line 300: the statement should be restricted to cognitive impairments tested in this manuscript, e.g.: "...Pyk2 deficit plays a significant role in hippocampal-dependent cognitive impairment in HD..."

4. Pg. 4, line 91: Should be "Shaffer" not Schaffer. Also on p. 11, line 308 and Fig. 1 legend, line 896.

5. What interval was used for the paired stimuli used to generate the PPR shown in Fig. 1e? This is not mentioned anywhere.

6. pg. 11, line 315-16: The citations (references 40 and 41) should both be moved to the end of the sentence since both papers focused on the role of Y1472 phosphorylation on synaptic (rather than just surface) localization of GluN2B.

REVIEWERS' COMMENTS:

The modified sentences are highlighted yellow in the revised text.

Reviewer #1 (Remarks to the Author):

The authors have done an excellent job of responding to the questions raised in the reviews, and have modified and edited the manuscript accordingly. They have even performed additional experiments that address the reviewers' concerns and surpass expectations. This is a timely and important report and is ready for publication. It was a pleasure to review such exceptional work.

We thank the Referee for these positive comments.

Reviewer #2 (Remarks to the Author):

The authors have addressed my concerns and the paper is suitable for publication.

We thank the Referee.

Reviewer #3 (Remarks to the Author):

The paper is significantly stronger after revisions, which have addressed almost all of my concerns. This is an impressive study of the role of Pyk2 in hippocampal CA1 synaptic structure and plasticity. My only remaining concern is the over-statement of a role for Pyk2 deficit in cognitive impairment in HD, which is detailed below along with a few minor corrections/suggestions.

1. The title should be modified. It is too strong to say that Pyk2 reduction in hippocampus (which is all that is studied in this paper) "mediates Huntington's disease cognitive deficits". The cognitive deficits in HD are numerous and diverse, and mainly involve frontal executive dysfunction, especially in early stages of disease. Results in this manuscript are restricted to demonstrating rescue of two hippocampal-dependent memory tasks in one mouse model of HD. Therefore, please soften the statement in the title; instead of "mediates" could say "contributes to".

We have modified the title as recommended.

2. Along the same lines as #1, on pg. 13, line 392 it is misleading to talk about the decrease in Pyk2 in Grade 3-4 HD brains as supporting a role for Pyk2 decrease in HD pathophysiology. The strongest data in this manuscript is from the R6/1 mouse model of HD, whereas the results from HD patient brains must be discussed in a more balanced way, mentioning that in Grade 3-4 HD brains there is significant synaptic loss so that many synaptic proteins show reductions; moreover, the fact that at earlier stages (Grade 1-2) there was no Pyk2 reduction suggests that the role of Pyk2 decrease in hippocampal-dependent cognitive deficits in HD patients remains to be determined.

We now include the following sentence: "In patients the role of Pyk2 deficiency remains to be determined since it was detected in Grade 3-4 when there is a loss of other synaptic proteins but not at earlier stages (Grade 1-2)."

3. Again, on pg. 11, line 300: the statement should be restricted to cognitive impairments tested in this manuscript, e.g.: "...Pyk2 deficit plays a significant role in hippocampal-dependent cognitive impairment in HD..."

The sentence has been modified to "We also provide evidence that Pyk2 deficit plays a significant role in hippocampal-dependent cognitive impairment in HD, a severe genetic neurodegenerative disorder."

4. Pg. 4, line 91: Should be "Shaffer" not Schaffer. Also on p. 11, line 308 and Fig. 1 legend, line 896.

This typo has been corrected. Thanks for pointing it out.

5. What interval was used for the paired stimuli used to generate the PPR shown in Fig. 1e? This is not mentioned anywhere.

The interval is now indicated in the legend to Figure 1e.

6. pg. 11, line 315-16: The citations (references 40 and 41) should both be moved to the end of the sentence since both papers focused on the role of Y1472 phosphorylation on synaptic (rather than just surface) localization of GluN2B.

We moved the references to the end of the sentence.